# ADAPTIVE CONSTRAINT INTEGRATION FOR SIMULTANEOUSLY OPTIMIZING CRYSTAL STRUCTURES WITH MULTIPLE TARGETED PROPERTIES

## ABSTRACT

In materials science, finding crystal structures that have targeted properties is crucial. While recent methodologies such as Bayesian optimization and deep generative models have made some advances on this issue, these methods often face difficulties in adaptively incorporating various constraints, such as electrical neutrality and targeted properties optimization, while keeping the desired specific crystal structure. To address these challenges, we have developed the Simultaneous Multi-property Optimization using Adaptive Crystal Synthesizer (SMOACS), which utilizes state-of-the-art property prediction models and their gradients to directly optimize input crystal structures for targeted properties simultaneously. SMOACS enables the integration of adaptive constraints into the optimization process without necessitating model retraining. Thanks to this feature, SMOACS has succeeded in simultaneously optimizing targeted properties while maintaining perovskite structures, even with models trained on diverse crystal types. We have demonstrated the band gap optimization while meeting a challenging constraint, that is, maintaining electrical neutrality in large atomic configurations up to 135 atom sites, where the verification of the electrical neutrality is challenging. The properties of the most promising materials have been confirmed by density functional theory calculations.

## 1 INTRODUCTION

We address the challenge of simultaneously optimizing multiple material properties while preserving specific crystal structures and ensuring electrical neutrality. To achieve this, we have developed a methodology that leverages property prediction models and their gradients to facilitate the discovery of materials with multiple desired properties. This approach allows for the adaptive application of constraints, such as electrical neutrality and specific crystal structures, without necessitating retraining. As a result, our method enables the optimization of large atomic configurations to obtain specific properties while ensuring electrical neutrality and preserving specific crystal structures.

Materials design is crucial for various advancing technologies, e.g., enhancing efficiency or reducing the cost of solar cells. The goal of materials design is to identify materials that simultaneously satisfy multiple property criteria, for instance, in terms of band gap and formation energy, while meeting other requirements, such as electrical neutrality. Furthermore, during the design process, it is often desirable to focus on specific promising systems, such as perovskite structures for next-generation solar cells (Green et al., 2014). In the exploration of specific crystal structures, elemental substitution—blending different elements—is commonly employed. For instance, blended perovskite structures might have alternative compositions such as $AA'BB'X_2X'X''_3$, which are derived from the standard $ABX_3$ format of a perovskite unit cell. Computational experiments involving these complex compositions often require larger systems that combine multiple unit cells. Consequently, the critical aspects of material design include 1) the ability to optimize multiple properties simultaneously, 2) the adaptive incorporation of various constraints, such as electrical neutrality or specific crystal structures, and 3) the ability to optimize large atomic configurations. In summary, we need to solve the problem of simultaneously optimizing multiple properties while preserving a specific crystal structure and ensuring electrical neutrality.

Advances in computational techniques have tremendously accelerated material design, with Density Functional Theory (DFT) becoming a standard tool for rapid property validation. Recent developments in machine learning have enabled faster property predictions through deep learning models trained on DFT-generated data. Models such as Crystalformer (Taniai et al., 2024), a transformer-based model (Vaswani et al., 2017), and ALIGNN (Choudhary & DeCost, 2021), a Graph Neural Network (GNN)-based model, significantly facilitate the screening process (Choubisa et al., 2023).

Bayesian optimization, such as Gaussian Process and Tree-structured Parzen estimator (TPE) (Watanabe, 2023), is commonly employed in material design (Ozaki et al., 2020a; Boyar et al., 2024; Zhai et al., 2024). A key advantage of Bayesian optimization is its capability to perform inverse inference, therefore enabling the prediction of crystal structures from given properties. Recently, deep generative models designed to synthesize crystal structures, such as FTCP (Ren et al., 2022), have gained much attention due to their potential to discover new stable materials. Additionally, there are methods that leverage large language models (Ding et al., 2024; Gruver et al., 2024), Generative Flow Networks (AI4Science et al., 2023), reinforcement learning (Govindarajan et al., 2024), or flow matching (Miller et al., 2024) to synthesize new crystal structures.

Despite these advances, many challenges remain. Firstly, research using deep generative models primarily aims to identify stable materials, and only a limited number of studies focus on optimizing both stability and key properties, such as the band gap, which is crucial for maximizing solar cell efficiency. Secondly, deep generative models are often built with specialized architectures, making it difficult to adopt the latest property prediction models for their prediction branches. This lack of flexibility in model architectures can hinder the improvement of prediction accuracy for generated materials. Thirdly, current generative models require retraining for targeted properties optimization within specific crystal structures, which is often the case in practice, such as perovskite structures for solar cells. Finally, verifying the electrical neutrality in large atomic configurations is complicated due to the combinatorial explosion resulting from the possible multiple oxidation numbers for many atomic species. Nevertheless, ensuring electrical neutrality is essential for proposing realistic materials.

To address these challenges, we have developed a framework, the **S**imultaneous **M**ulti-property **O**ptimization using **A**daptive **C**rystal **S**ynthesizer (SMOACS). SMOACS can employ various property prediction models as far as their gradients can be computed and optimizes input crystal structures directly to achieve target properties through the backpropagation technique (Fig. 1(left)). This approach enables accurate prediction of multiple properties and simultaneous optimization by utilizing several recently developed pre-trained models for predicting different material properties. Unlike methods using normalizing flows that require architectural constraints for invertibility, our method imposes no such restrictions on these models. When newer models become available in the future, improved prediction accuracy will be achieved by incorporating them into our approach. Additionally, by managing the optimization range and utilizing special loss functions, we facilitate targeted properties optimization within specific crystal structures, avoiding retraining. Moreover, by imposing constraints via combinations of oxidation numbers, our method ensures the electrical neutrality of any proposed materials, even in large atomic configurations where verifying electrical neutrality is difficult due to combinatorial explosion. The generalizability of SMOACS enables it to adopt various prediction models and optimize various properties.

SMOACS is the first method that directly optimizes the space of crystal structures using a gradient-based approach. We achieve this by making the entire crystal structure differentiable, which involves decomposing it into various components and representing atomic species as atomic distributions. Unlike traditional methods that convert crystal structures into latent variables (Ren et al., 2022)—thereby entangling their elements—our approach maintains the independence of each component. This independence facilitates the preservation of crystal structures and ensures electrical neutrality by precisely specifying the atoms at each site. Furthermore, unlike generative models that probabilistically generate materials satisfying certain conditions, our method can inherently guarantee electrical neutrality and the preservation of crystal structures. Moreover we can add additional constraints as long as they are differentiable.

We demonstrated that SMOACS could effectively utilize both GNN-based models and transformer-based models, outperforming FTCP, deep generative models, and TPE, Bayesian optimization. We demonstrated the band gap optimization within perovskite structures without retraining, using models trained on the MEGNet dataset (Chen et al., 2019), which includes various types of crystals.

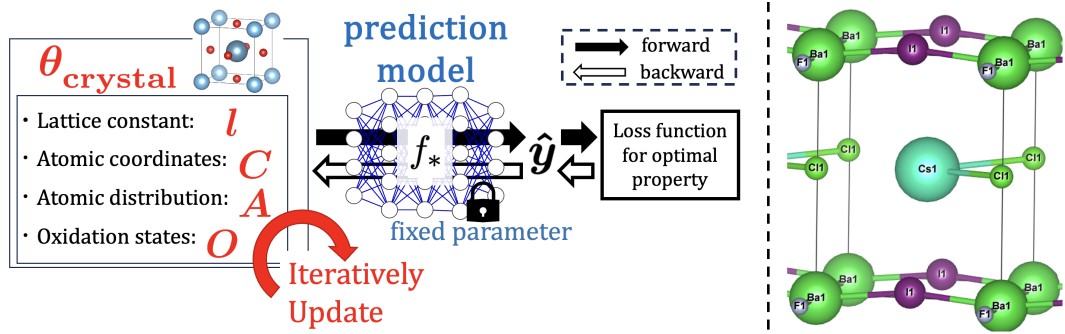

Figure 1: (left) Overview of the SMOACS framework. (right) An example of an optimized perovskite structure with a 4.02 eV band gap, verified at 3.96 eV through DFT calculations. Visualization was done with VESTA (Momma & Izumi, 2011).

Additionally, we demonstrated the optimization for large atomic configurations with as many as 135 atom sites while ensuring electrical neutrality. Furthermore, the validity of the proposed materials was verified through DFT calculations.

## 2    RELATED WORKS

**Property prediction model**. In recent years, much research has actively focused on predicting the properties of materials using DFT-generated data (Davariashtiyani & Kadkhodaei, 2023; Merchant et al., 2023; Yang et al., 2024a). There are two primary approaches involving deep learning. The first approach utilizes GNNs (Chen et al., 2019; Park & Wolverton, 2020; Louis et al., 2020; Schmidt et al., 2021; Lin et al., 2023), such as ALIGNN. The main advantage of using GNNs is their ability to graphically represent crystal structures, thereby considering inter-atomic relationships in more physically meaningful ways. The second approach employs transformers (Ying et al., 2021; Yan et al., 2022), such as Crystalformer, which are known for their promising performance in the field of computer vision and natural language processing (Brown et al., 2020; Dosovitskiy et al., 2021).

**Deep generative models**. Deep generative models, including language models, for producing new stable materials have been emerging in the last years (Xie et al., 2022; Lyngby & Thygesen, 2022; Sultanov et al., 2023; Yang & Mannodi-Kanakkithodi, 2022). However, only a few studies explored material properties and stability at the same time. Studies such as FTCP (Ren et al., 2022), MatterGen (Zeni et al., 2023), and UniMat (Yang et al., 2024b) focused on optimizing properties including band gap and material stability. They are generative models and thus conduct property optimizations within the framework of generative modeling. FTCP, based on Variational Autoencoders (Kingma, 2013), encodes crystal structures into latent variables. It employs prediction branches to predict properties from these variables. MatterGen and UniMat are diffusion models and employ classifier-free guidance (Ho & Salimans, 2021) to generate materials with specific properties. Although methods exist to constrain condition-free models for generating specific outputs (Wu et al., 2024), no research has implemented these techniques for crystal structures.

**Bayesian Optimization**. Black-box optimization, including Bayesian optimization, is widely used in materials science (Song et al., 2024). Numerous studies in materials science apply Bayesian optimization to predict crystal and molecular structures from target properties (Boyar et al., 2024; Zhai et al., 2024; Khatamsaz et al., 2023). One representative method of Bayesian optimization widely utilized in materials science research is the Gaussian process (GP) (Lu et al., 2022). However, as the Gaussian process only handles continuous values, its ability to manage categorical variables like elements is questionable. The recently proposed Tree-structured Parzen Estimator (TPE), which is capable of handling categorical variables and multi-objective optimization (Ozaki et al., 2020b; 2022) and has been utilized in materials science (Ozaki et al., 2020a), could be a better choice.

**Gradient based approach**. Gradient-based approaches that aim to optimize design variables toward desired properties using deep learning-based predictors and their gradients have been applied across a wide range of fields. For example, they have been used to optimize designs for dynamics of

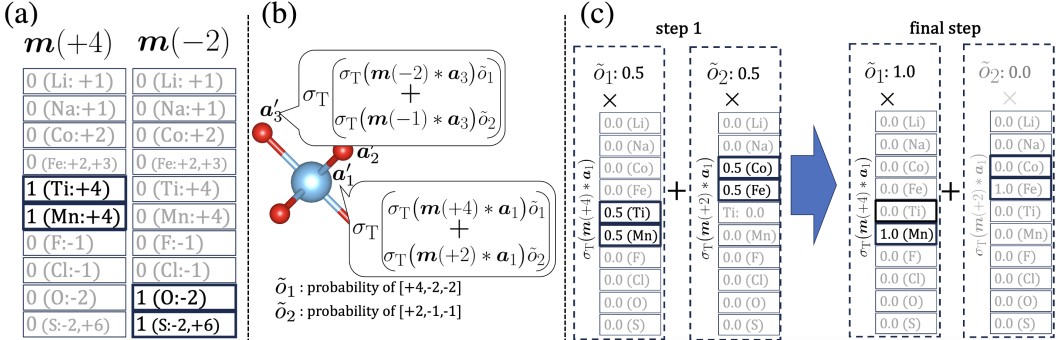

Figure 2: (a) SMOACS enforces site-specific restrictions on the types of elements to maintain electrical neutrality. The oxidation masks, labeled $\boldsymbol{m}(+4)$ and $\boldsymbol{m}(-2)$, correspond to elements with oxidation numbers of $+4$ and $-2$, respectively. The values 0 and 1 indicate the values of the mask. Parentheses indicate the elements and their potential oxidation numbers at each position. (b) Atomic distributions at each site considering two potential oxidation numbers. Here, we consider two possible patterns of oxidation number combinations: $[+4, -2, -2]$ and $[+2, -1, -1]$ for three sites. The atomic distributions at each site are computed by taking the weighted sum of the probabilities of these patterns. (c) Change in the atomic distribution at site No.1 in the optimization process. The numbers in the grids indicate the probabilities of elements in parentheses. As a result of the optimization, the $\mathrm{TiO_2}$-type oxidation pattern $[+4, -2, -2]$ is selected, with $\mathrm{Mn}$ chosen as the element that achieves a $+4$ oxidation number.

physical systems (Allen et al., 2022; Hwang et al., 2022), image manipulation (Xia et al., 2022), metamaterials (Bordiga et al., 2024), and chemical compositions (Fujii et al., 2024) to achieve target performance. These methods require that the chain rule of differentiation connects from the input to the output. While there is the study that apply this technique by mapping crystal structures into latent spaces (Xie et al., 2022), there are no studies that apply it directly within the space of crystal structures.

## 3 SMOACS

In SMOACS, the crystal structure $\boldsymbol{\theta_{\mathrm{crystal}}}$ is divided into four learnable parameters: lattice constant $\boldsymbol{l}$, coordinates of $N$ atomic sites $\boldsymbol{C}$, elements $\boldsymbol{e}$, and an oxidation state configuration parameter $\boldsymbol{o}$ (Fig. 1(left)).

$$\boldsymbol{\theta_{\mathrm{crystal}}} = \{\boldsymbol{l}, \boldsymbol{C}, \boldsymbol{e}, \boldsymbol{o}\} \tag{1}$$

$$\boldsymbol{l} \in \mathbb{R}^6, \ \ \boldsymbol{C} \in \mathbb{R}^{N \times 3}, \ \ \boldsymbol{e} \in \mathbb{R}^N, \ \ \boldsymbol{o} \in \mathbb{R}^D \tag{2}$$

The lattice constant $\boldsymbol{l}$ comprises the crystallographic axes lengths $a, b, c$ and the angles between these axes $\alpha, \beta, \gamma$. The oxidation state configuration parameter $\boldsymbol{o}$ denotes the probabilities for D patterns of oxidation number combinations determined by initial crystal structures, further described in Section 3.1. The $\boldsymbol{l}$ and $\boldsymbol{C}$, being continuous variables, can be optimized directly through backpropagation technique (Ren et al., 2020; Fujii et al., 2023). However, this technique cannot be used for the elements $\boldsymbol{e}$ since they are being discrete and categorical. Therefore, instead of directly handling the elements $\boldsymbol{e}$, we employ a technique where an element at site $n$ is represented by the atomic distribution $\boldsymbol{a}_n$ ($\boldsymbol{a}_n \in \mathbb{R}^K$, $\boldsymbol{A} \in \mathbb{R}^{N \times K}$, $\boldsymbol{A}_{i,:} = (\boldsymbol{a_i})^\top$) (Konno et al., 2021; Fujii et al., 2024). Here, $K$ represents the highest atomic number considered. Since we are dealing with atomic numbers from 1 to 98, $K = 98$. When an element with the atomic number $k$ occupies site $n$, $\boldsymbol{a}_n$ becomes a one-hot vector with the element $k$ set to 1 and 0 at all others. Please refer to Section A.5 for a discussion on the general applicability of using atomic distributions in various property prediction models.

### 3.1 MASKS TO MAINTAIN ELECTRICAL NEUTRALITY

To maintain electrical neutrality, we restrict the possible values of atomic distribution $\boldsymbol{a}_n$ by using a mask that aligns with the possible oxidation numbers at site $n$. These possible oxidation numbers are determined from the initial structure. Here, we explain this using the rutile type structure as an example. A typical material having this structure is titanium dioxide ($TiO_2$). The rutile $TiO_2$ contains one Ti site and two O sites, totaling three atomic sites. When titanium has an oxidation number of $+4$ and the two oxygen atoms each have an oxidation number of $-2$, the total oxidation number is zero, thus achieving electrical neutrality. Therefore, to maintain electrical neutrality, we can use an atomic distribution that includes only elements with a $+4$ oxidation number, such as Ti and Mn at the Ti site. At the O sites, we use those with an oxidation number of $-2$, such as O and S. This ensures electrical neutrality regardless of the elements selected after optimization. The adjusted atomic distribution $\boldsymbol{a}'_n$, which considers oxidation numbers, is obtained by taking the element-wise product of the learnable distribution $\boldsymbol{a}_n$ with the atomic mask $\boldsymbol{m}(s)$.

$$\boldsymbol{a}'_n(s) = \sigma(\boldsymbol{m}(s) * \boldsymbol{a}_n) \tag{3}$$

Here, $\boldsymbol{m}(s)$ is $\boldsymbol{m}(s) \in \mathbb{R}^K$ and a mask that assigns a value of 1 to elements with the oxidation number $s$, and 0 to all others (Fig. 2(a)). $\sigma$ is a normalization function that rescales all elements to the range [0,1], with their total sum normalized to 1.0. The asterisk denotes element-wise multiplication. This process is applied to all sites, yielding an atomic distribution $\boldsymbol{A}'(\boldsymbol{S})$ that reflects the oxidation numbers for all sites.

$$\boldsymbol{A}'(\boldsymbol{S}) = \sigma_{\text{atom}}(\boldsymbol{A} * \boldsymbol{M}(\boldsymbol{S})) \tag{4}$$

$$\boldsymbol{M} \in \mathbb{R}^{N \times K}, \ \boldsymbol{M}(\boldsymbol{S})_{i,:} = (\boldsymbol{m}(s_i))^\top, \ \boldsymbol{S}_i = s_i, \ \boldsymbol{m}(s_i) \in \{\boldsymbol{m}(s_{\min}), ..., \boldsymbol{m}(s), ..., \boldsymbol{m}(s_{\max})\} \tag{5}$$

Here, $\sigma_{\text{atom}}$ is a function that normalizes values along elemental directions. $s_i$ is the oxidation number at site-$i$. The $s_{\min}$ and $s_{\max}$ respectively denote the minimum and maximum oxidation numbers among all elements considered. To simultaneously consider different patterns of oxidation number combinations, we introduce a learnable parameter $\boldsymbol{o}$, which selects the optimal combination of oxidation numbers. We illustrate this approach using $CoF_2$ and $TiO_2$, both of which adopt the rutile structure. The oxidation numbers differ at each atomic site, with $CoF_2$ exhibiting oxidation numbers of $[+2, -1, -1]$ and $TiO_2$ having $[+4, -2, -2]$. The $\boldsymbol{o}$ is a $d$-dimensional vector selecting the best pattern from $d$ patterns of oxidation number combinations. The $\boldsymbol{o}$ represents the probabilities of each combination. For instance, when considering two patterns, such as those of $CoF_2$ and $TiO_2$, $d$ equals 2. Using this framework, we can calculate the modified atomic distribution $\boldsymbol{a}'_n$ (Fig. 2(b)) considering multiple combination patterns as follows:

$$\boldsymbol{a}'_n(\boldsymbol{a}_n, o_d) = \sum_{d=1}^{D} \boldsymbol{a}'_{n,d} = \sum_{d=1}^{D} \sigma(\boldsymbol{m}(s_{n,d}) * \boldsymbol{a}_n)o_d. \tag{6}$$

Here, $\boldsymbol{m}(s_{i,d}) \in \{\boldsymbol{m}(s_{\min}), ..., \boldsymbol{m}(s), ..., \boldsymbol{m}(s_{\min})\}$. The property prediction models assume that each site contains a single element. Therefore, after optimization, it is desirable for the oxidation state configuration parameter $\boldsymbol{o}$ and atomic distributions $\boldsymbol{a}'_n$ to become one-hot vectors. To guarantee that optimization will result in them becoming one-hot vectors, we normalize $\boldsymbol{a}'_n$ and $\boldsymbol{o}$ with the temperature softmax function $\sigma_T$.

$$\sigma_T(z_i) = \frac{\exp\left(\frac{z_i}{T}\right)}{\sum_j \exp\left(\frac{z_j}{T}\right)} \tag{7}$$

$$\tilde{\boldsymbol{o}} = \sigma_T(\boldsymbol{o}) \tag{8}$$

$$\tilde{\boldsymbol{A}}_n\left(\boldsymbol{A}_n, \boldsymbol{o}, T\right) = \sigma_T\left(\sum_{d=1}^{D} \boldsymbol{A}'_d(\boldsymbol{S}_d)\right) = \sigma_{T,\text{atom}}\left(\sum_{d=1}^{D} \sigma_{T,\text{atom}}(\boldsymbol{M}(\boldsymbol{S}_d) * \boldsymbol{A})\tilde{o}_d.\right) \tag{9}$$

$$(\boldsymbol{M}(\boldsymbol{S}_d) \in \mathbb{R}^{N \times K}, \ \boldsymbol{M}(\boldsymbol{S}_d)_{i,:} = \boldsymbol{m}(\boldsymbol{s}_{i,d})^\top, \ (\boldsymbol{S}_d)_i = s_{i,d} \tag{10}$$

Here, $\boldsymbol{M}(\boldsymbol{S}_d)$ is an atomic mask of the $d$-th oxidation pattern. $\sigma_T$ produces sharper distributions at lower temperatures $T$, ensuring that the parameters transition into one-hot vectors. For example, let $o_1$ represent the probability of the $TiO_2$ type with oxidation numbers [+4, -2, -2], and $o_2$ represent the $CoF_2$ type with oxidation numbers [+2, -1, -1]. If, at the end of optimization, $\boldsymbol{o} = (1., 0.)$ is achieved, the $TiO_2$ type is selected, resulting in a material with an oxidation pattern of [+4, -2, -2], as shown in Fig. 2(c).

## 3.2 Initialization and Multiple Properties Optimization

In SMOACS, the initial structures for the optimization process are obtained from two sources: an existing dataset and self-generated structures based on typical perovskites with modified lattice constants (for details, see Section A.6 and A.7). These crystal structures must satisfy electrical neutrality and generate $D$ oxidation number patterns based on the compositions of initial crystal structures (see Section A.11 for details). The lattice constant $\boldsymbol{l}$ and atomic coordinates $\boldsymbol{C}$ are used directly as initial values. The atomic distribution $\boldsymbol{A}$ and the oxidation number pattern selection parameter $\boldsymbol{o}$ are initialized with a uniform distribution.

$$\theta'_{\text{crystal}} = \tau\Big(\{\boldsymbol{l}, \boldsymbol{C}, \tilde{\boldsymbol{A}}(\boldsymbol{A}, \boldsymbol{o}, T), \tilde{\boldsymbol{o}}(\boldsymbol{o}, T)\}\Big) \tag{11}$$

$$\boldsymbol{l} \leftarrow -\eta_l \frac{\partial L}{\partial \boldsymbol{l}}, \quad \boldsymbol{C} \leftarrow -\eta_C \frac{\partial L}{\partial \boldsymbol{C}}, \quad \boldsymbol{A} \leftarrow -\eta_A \frac{\partial L}{\partial \boldsymbol{A}}, \quad \boldsymbol{o} \leftarrow -\eta_o \frac{\partial L}{\partial \boldsymbol{o}}. \tag{12}$$

Here, $\eta_l, \eta_C, \eta_A, \eta_o$ denote the learning rates for each parameter. L denotes loss function $L\big(f_*(\boldsymbol{\theta}'_{\textbf{crystal}}), \boldsymbol{y}_{\text{target}}\big)$ and $f_*$ denotes a set of trained models. The $\tau$ is a function converting structures to inputs for $f_*$. We optimize structures by iteratively updating them using Eq. 11 and 12. During optimization, the temperature $T$ of the softmax function starts high and is lowered towards the end, forcing $\tilde{\boldsymbol{o}}$ and $\tilde{\boldsymbol{a}}_n$ into one-hot vectors in the final stage. SMOACS optimizes multiple properties by incorporating various trained models or additional loss functions. Here, we aim to optimize the crystal structure by minimizing formation energy and targeting a specific band gap range, $y_{\text{bg}} \pm h_{\text{bg}}$, using trained models $f_* = \{f_{\text{bg}*}, f_{\text{f}*}\}$. Here, $h_{\text{bg}}$ is an acceptable margin and $f_{\text{bg}*}$ is the trained model predicting the band gap, and $f_{\text{f}*}$ predicts the formation energy. We also set a strength parameter $\lambda$.

$$L_{\text{bg}}(y_{\text{bg}}, \hat{y}_{\text{bg}}) = \max(0, |y_{\text{bg}} - \hat{y}_{\text{bg}}| - h_{\text{bg}}), \quad L_{\text{f}}(\hat{y}_{\text{f}}) = \hat{y}_{\text{f}} \tag{13}$$

$$L = L_{\text{bg}}(y_{\text{bg}}, f_{\text{bg}*}(\theta'_{\text{crystal}})) + \lambda L_{\text{f}}(f_{\text{f}*}(\theta'_{\text{crystal}})) \tag{14}$$

The influence of $\lambda$ is discussed in Section A.12. Note that since the crystal structure changes during optimization, when using GNNs, we update the graph multiple times based on the current structure in the optimization process.

## 3.3 Preservation of Specific Crystal Structures during Optimization

Limiting the optimization variables and their range allows us to maintain specific crystal structures during optimization. For instance, let us consider a typical perovskite structure, represented by the chemical formula $ABX_3$. It consists of five sites and adopts a crystal structure close to a cubic lattice. The fractional coordinates for the five sites are as follows: $(0.5, 0.5, 0.5)$ at the A site, $(0.0, 0.0, 0.0)$ at the B site, and $(0.5, 0.0, 0.0)$, $(0.0, 0.5, 0.0)$, $(0.0, 0.0, 0.5)$ at the three X sites. Note that deviations from these values are allowed, together with degrees of freedom related to the lattice constant values. We optimize the structures within the range of small perturbations applied to typical perovskites. Specifically, first, only the $a, b,$ and $c$ of $\boldsymbol{l}$ are optimized, while $\alpha, \beta,$ and $\gamma$ are fixed at $90°$. Subsequently, the optimization range for the five sites is set close to their typical coordinates. For example, we optimize the coordinates at the A site within the range $(0.5 \pm \epsilon, 0.5 \pm \epsilon, 0.5 \pm \epsilon)$, where $\epsilon$ is a small constant. Following a previous work on the distortion of $CaCu_3Ti_4O_{12}$ (Božin et al., 2004), we set $\epsilon = 0.15$. We also specify possible patterns of oxidation number combinations. Typically, some materials with perovskite structure such as $SrTiO_3$ exhibit oxidation numbers of $[+2, +4, -2]$ at the A, B, and X sites, respectively, while others such as $(CH_3NH_3)PbI_3$ exhibit $[+1, +2, -1]$. Consequently, two oxidation number patterns are prepared for the perovskite structure: $[+2, +4, -2]$ and $[+1, +2, -1]$ for the A, B, and X sites, respectively. By specifying these variables and ranges for optimization, we are able to maintain the perovskite structure.

## 4 EXPERIMENTS

We compare SMOACS's ability to propose material satisfying specified properties and constraints with those from deep generative models and Bayesian optimization, represented by FTCP and TPE, respectively. TPE was chosen over GP, as discussed in Section 2. Both SMOACS and TPE ran for 200 optimization steps. All models were trained on the MEGNet dataset. For further implementation details, please refer to Section A.4. To demonstrate SMOACS's versatility across various property prediction models, we conducted optimizations using ALIGNN and Crystalformer, GNN-based and transformer-based models, respectively. We evaluated the optimized materials using three metrics: whether they satisfied the specified criteria on the band gap range, formation energy, and validity of crystal structure. Specifically, we judge that the formation energy criterion is satisfied if it is less than -0.5 eV, and for the validity of crystal structure, following a previous research (Xie et al., 2022), we adopted two criteria: all interatomic distances being at least 0.5 Å and maintaining electrical neutrality. For assessing electrical neutrality, we consider the material electrically neutral if the sum of possible oxidation numbers for atoms at each site equals zero. Please refer to Section A.9 for details. For experiments utilizing models trained on datasets other than MEGNet, please refer to Section A.13.

### 4.1 LEVERAGING THE LATEST RESEARCH ACHIEVEMENTS IN PROPERTY PREDICTION

First, we experimented with the performance of property prediction models that could be adopted in systems such as SMOACS, Bayesian optimization (TPE), or FTCP. In principle, Bayesian optimization—a type of black-box optimization—and SMOACS can adopt a broad range of property prediction models. Meanwhile, FTCP—a generative model—employs a property prediction branch within its architecture. Therefore, unlike SMOACS and Bayesian optimization, FTCP cannot use ALIGNN or Crystalformer for property predictions. The results are shown in Table 1.

Crystalformer demonstrated the highest prediction accuracy among the models in Table 1. ALIGNN ranked second, whereas the prediction branches of FTCP exhibited the lowest performance. This result confirmed an advantage of SMOACS and TPE: their ability to incorporate state-of-the-art property prediction models, such as Crystalformer and ALIGNN, allowing for highly accurate material property predictions.

### 4.2 SIMULTANEOUS OPTIMIZATION OF TARGETED PROPERTIES REGARDLESS OF THE CRYSTAL STRUCTURE

We tested the ability to optimize band gaps to target values. In this experiment, we optimized both the band gap and formation energy simultaneously, regardless of the crystal structure. To ensure a fair comparison of optimization methods, we fixed the margin for all band gaps at $\pm 0.04$ eV. For optimization results where the predictor's error is used as the margin, please refer to Section A.14. We conducted experiments with SMOACS using Crystalformer and ALIGNN, respectively. We utilized Crystalformer as a predictor for TPE. We used three objective functions for TPE: band gap, formation energy, and electrical neutrality. FTCP selected data from the MEGNet dataset close to the target band gap and with formation energy less than $-0.5$ eV, subsequently encoding them into latent variables. Finally, after adding noise, we decoded them back into crystal structures for evaluation. We optimized and evaluated the structures based on the band gap and formation energy values predicted by their respective predictors.

Table 1: Comparison of property prediction models. This table compares models trained on the MEGNet dataset and presents Mean Absolute Error (MAE) scores for formation energy (**E_form**) and band gap on test data in the MEGNet dataset. Lower scores are better across all metrics.

| Prediction Model | Applicable Method | E_form MAE (eV) | Band Gap MAE (eV) |
|---|---|---|---|
| Prediction Branch of FTCP | FTCP | 0.224 | 0.442 |
| ALIGNN | SMOACS, TPE | 0.022 | 0.218 |
| Crystalformer | SMOACS, TPE | **0.019** | **0.198** |

Table 2: Experiments on optimizing band gaps. We define the success rate as the probability of simultaneously satisfying three conditions: (A) the band gap is optimized within the target range, (B) the formation energy is below $-0.5$ eV, and (C) the crystal structure is valid. C is achieved when two criteria are met simultaneously: (a) all inter-atomic distances are greater than 0.5 Å, and (b) the structure is electrically neutral. S(Cry) and S(ALI) denote SMOACS utilizing the Crystalformer and ALIGNN models, respectively. We evaluated each of the proposed materials using all evaluation metrics, and the results were averaged over 256 samples. Higher scores are better across all metrics. Augmented results are shown in Table A.2.

| Target BG (eV) | method | success rate | (A)BG | (B)$E_f$ | (C)STR | (a) neut | (b) 0.5Å |
|---|---|---|---|---|---|---|---|
| 0.50 ±0.04 | S(Cry) | **0.328** | 0.465 | 0.566 | 0.758 | 0.957 | 0.758 |
| | S(ALI) | 0.055 | 0.062 | 0.867 | 0.867 | 0.949 | 0.867 |
| | TPE | 0.004 | 0.945 | 0.059 | 0.066 | 0.070 | 0.910 |
| | FTCP | 0.000 | 0.004 | 1.000 | 0.719 | 0.746 | 0.906 |
| 1.50 ±0.04 | S(Cry) | **0.387** | 0.543 | 0.672 | 0.824 | 0.980 | 0.824 |
| | S(ALI) | 0.043 | 0.066 | 0.828 | 0.852 | 0.938 | 0.852 |
| | TPE | 0.020 | 0.855 | 0.055 | 0.074 | 0.082 | 0.828 |
| | FTCP | 0.000 | 0.000 | 1.000 | 0.703 | 0.723 | 0.895 |
| 2.50 ±0.04 | S(Cry) | **0.383** | 0.473 | 0.715 | 0.840 | 0.984 | 0.840 |
| | S(ALI) | 0.051 | 0.059 | 0.809 | 0.793 | 0.898 | 0.793 |
| | TPE | 0.023 | 0.711 | 0.098 | 0.051 | 0.055 | 0.816 |
| | FTCP | 0.004 | 0.004 | 1.000 | 0.695 | 0.707 | 0.902 |

The results are shown in Table 2. SMOACS significantly outperformed both TPE and FTCP in terms of success rates. SMOACS consistently maintained electrical neutrality, except for extreme geometries causing NaN values during crystal vector calculations. While FTCP always met the requirements for formation energy, it struggled to achieve the target band gap, contributing to its lower overall success rate. TPE achieved a high success rate in optimizing the band gap within the target range, but it could not optimize formation energy well. SMOACS maintained a high overall success rate as it achieved substantial success rates in both band gap and formation energy optimization. SMAOCS can easily scale this computation and can optimize 2,048 samples simultaneously in just a few minutes using a single A100 GPU. This allows us to repeat the optimization process multiple times, enabling us to obtain a large number of successful optimization samples. Please refer to Section A.6 for details, including the diversity of generated materials.

## 4.3 SIMULTANEOUS OPTIMIZATION OF TARGETED PROPERTIES WHILE PRESERVING PEROVSKITE STRUCTURES

We optimized the band gap within the range of 0.5 to 2.5 eV while preserving the perovskite structure. Besides the previously discussed metrics, we used three new criteria to confirm a structure's perovskite identity: internal coordinates, angles between crystal axes, and the tolerance factor. The tolerance factor $t$ serves as a metric to assess the suitability of atomic combinations for forming perovskite structures (West, 2022). $t$ is calculated based on the ionic radii $r_A$, $r_B$ and $r_X$ of each site in the perovskite structure $ABX_3$ and we employed a loss function to optimize the tolerance factor alongside minimizing the band gap and formation energy.

$$t = \frac{r_A + r_X}{\sqrt{2}(r_B + r_X)} \tag{15}$$

$$L_t = |t - 0.9| \tag{16}$$

$$L = L_{bg}(y_{bg}, f_{bg*}(\theta'_{crystal})) + L_f(f_{f*}(\theta'_{crystal})) + L_t(\theta'_{crystal}) \tag{17}$$

If $t$ is close to 1, the structure is likely a perovskite; if it is far from 1, it is not. Considering $t$ values of typical perovskite structures ($BaCeO_3$:0.857, $SrTiO_3$: 0.910 and $BaTiO_3$: 0.970), we established a tolerance factor range of $0.8 \leq t \leq 1.0$ as the criterion for success. SMOACS optimized the structures with the procedure outlined in Section 3.3. Due to the limited number of perovskite

Table 3: Experiments on optimizing various band gaps while preserving perovskite structures. The "success rate" is the probability of simultaneously satisfying four criteria: (A) the band gap is optimized within the target range, (B) the formation energy is below $-0.5$ eV, (C) the crystal structure is valid, and (D) approximating a valid perovskite structure. Criteria (A), (B), and (C) are consistent with those outlined in Table 2. The (D) is achieved when three criteria are met simultaneously: (c) the tolerance factor $t$ is between 0.8 and 1.0, (d) coordinates are within $\pm 0.15$ of typical perovskite structure coordinates, and (e) crystal axis angles are from $85°$ to $95°$. The results are averaged over 256 samples. Higher scores are better across all metrics. Augmented results are shown in Table A.4.

| Target BG (eV) | method | success rate | (A)BG | (B)$E_f$ | (C)STR | (a) neut | (b) 0.5Å | (D)PS | (c) tole | (d) angles | (e) coord |
|---|---|---|---|---|---|---|---|---|---|---|---|
| 0.50 ±0.04 | S(Cry) | **0.113** | 0.477 | 0.410 | 0.965 | 1.000 | 0.965 | 0.500 | 0.500 | 1.000 | 1.000 |
| | S(ALI) | 0.090 | 0.211 | 0.535 | 1.000 | 1.000 | 1.000 | 0.500 | 0.500 | 1.000 | 1.000 |
| | TPE | 0.027 | 1.000 | 0.137 | 0.535 | 0.535 | 1.000 | 0.648 | 0.648 | 1.000 | 1.000 |
| | FTCP | 0.004 | 0.023 | 1.000 | 0.836 | 0.840 | 0.938 | 0.258 | 0.508 | 0.441 | 0.285 |
| 1.50 ±0.04 | S(Cry) | **0.148** | 0.422 | 0.461 | 0.984 | 1.000 | 0.984 | 0.578 | 0.578 | 1.000 | 1.000 |
| | S(ALI) | 0.070 | 0.219 | 0.652 | 1.000 | 1.000 | 1.000 | 0.629 | 0.629 | 1.000 | 1.000 |
| | TPE | 0.023 | 0.992 | 0.281 | 0.293 | 0.293 | 1.000 | 0.523 | 0.523 | 1.000 | 1.000 |
| | FTCP | 0.000 | 0.016 | 1.000 | 0.895 | 0.906 | 0.965 | 0.242 | 0.547 | 0.418 | 0.320 |
| 2.50 ±0.04 | S(Cry) | **0.152** | 0.285 | 0.516 | 0.988 | 1.000 | 0.988 | 0.625 | 0.625 | 1.000 | 1.000 |
| | S(ALI) | 0.113 | 0.184 | 0.938 | 1.000 | 1.000 | 1.000 | 0.625 | 0.625 | 1.000 | 1.000 |
| | TPE | 0.016 | 0.918 | 0.281 | 0.352 | 0.352 | 1.000 | 0.387 | 0.387 | 1.000 | 1.000 |
| | FTCP | 0.008 | 0.012 | 0.996 | 0.879 | 0.898 | 0.953 | 0.250 | 0.543 | 0.441 | 0.289 |

structures in the MEGNet dataset, random perovskite configurations are used as initial values for SMOACS and TPE. The optimization range for SMOACS and TPE is established as outlined in Section 3.3. FTCP initially encoded typical perovskite structures from the MEGNet dataset into latent variables. After adding noise, these latent variables are decoded back into crystal structures for evaluation. Please refer to Section A.7 for details, including the diversity analysis. We evaluated the structures based on the band gap and formation energy values predicted by their respective predictors. The evaluation results are shown in Table 3.

SMOACS with Crystalformer significantly outperformed both TPE and FTCP in overall success rates while preserving perovskite structures. In terms of (d) coordinates and (e) angles, both SMOACS and TPE consistently meet the criteria because their optimization ranges are the same. The generative model (FTCP), which uses latent variables, fails to obtain specific structural features of perovskite. Note that this limitation occurs despite the use of latent variables based on typical perovskite structures. This seems to be attributed to the training dataset that includes mixed crystal types. It is noteworthy that SMOACS consistently ensures electrical neutrality.

## 4.4 OPTIMIZING LARGE ATOMIC CONFIGURATIONS

We optimized large atomic configurations where calculating electrical neutrality is impractical. In systems containing many atoms, the calculation of electrical neutrality becomes infeasible due to combinatorial explosion. For example, a system containing 135 atoms, each with two possible oxidation numbers, results in about $4.3 \times 10^{40}$ combinations. Therefore, including an objective function for electrical neutrality in the TPE is infeasible. We conducted experiments on $3 \times 3 \times 3$ perovskite structures containing 135 atom sites and compared SMOACS with TPE, not including the objective function for electrical neutrality (referred to as TPE(/N)). The results are shown in Table 4.

SMOACS successfully optimized large atomic configurations, while TPE(/N) failed due to its inability to optimize the formation energy. The success of SMOACS likely stems from the utilization of information on gradients to optimize based on physics, enabling optimization even in large and complex systems. Furthermore, TPE was not able to evaluate electrical neutrality due to the computational cost of calculating it. Conversely, since SMOACS always maintains electrical neutrality, it is able to optimize properties while preserving this neutrality. Please refer to Section A.8 for details.

Table 4: Experiments optimizing for various band gaps while preserving $3 \times 3 \times 3$ perovskite structures. We include only TPE, showing better performance in Section 4.3, for comparison. Evaluation methods are based on those described in Table 3. Augmented results are shown in Table A.6.

| Target BG (eV) | method | success rate | (A)BG | (B)E$_f$ | (C)STR | (a) neut | (b) 0.5Å | (D)PS | (c) tole | (d) angles | (e) coord |
|---|---|---|---|---|---|---|---|---|---|---|---|
| 0.50 ±0.04 | S(Cry) | 0.156 | 0.734 | 0.547 | 0.968 | 1.00 | 0.969 | 0.570 | 0.570 | 1.000 | 1.000 |
| | S(ALI) | **0.188** | 0.234 | 0.812 | 0.687 | 1.00 | 0.688 | 0.789 | 0.789 | 1.000 | 1.000 |
| | TPE(/N) | 0.000 | 1.000 | 0.000 | - | N/A | 1.000 | 0.609 | 0.609 | 1.000 | 1.000 |
| 1.50 ±0.04 | S(Cry) | 0.047 | 0.125 | 0.422 | 0.953 | 1.00 | 0.953 | 0.617 | 0.617 | 1.000 | 1.000 |
| | S(ALI) | **0.062** | 0.086 | 0.867 | 0.726 | 1.00 | 0.727 | 0.586 | 0.586 | 1.000 | 1.000 |
| | TPE(/N) | 0.000 | 0.141 | 0.000 | - | N/A | 1.000 | 0.180 | 0.180 | 1.000 | 1.000 |
| 2.50 ±0.04 | S(Cry) | 0.023 | 0.039 | 0.438 | 0.984 | 1.00 | 0.984 | 0.664 | 0.664 | 1.000 | 1.000 |
| | S(ALI) | **0.102** | 0.172 | 1.000 | 0.703 | 1.00 | 0.703 | 0.812 | 0.812 | 1.000 | 1.000 |
| | TPE(/N) | 0.000 | 0.023 | 0.000 | - | N/A | 0.984 | 0.156 | 0.156 | 1.000 | 1.000 |

## 4.5 VERIFICATION BY DENSITY FUNCTIONAL THEORY

We used Density Functional Theory (see Section A.1 for details) to verify the band gaps of materials proposed by SMOACS. Among these materials, $BaCsFClI$ (Fig. 1(right)), a perovskite structured for a 4.02 eV band gap, showed a DFT-calculated value of 3.96 eV. However, we also found discrepancies between the values the model predicted and the DFT calculated for other candidate materials. Detailed results can be found in the appendix, Section A.2.

## 5 CONCLUSIONS

We propose SMOACS, a framework that utilizes the latest high-accuracy property prediction models and their gradients to search for materials with targeted multiple properties. SMOACS can adaptively apply constraints such as electrical neutrality and specific crystal structures without retraining. SMOACS not only outperformed FTCP and TPE in optimizing multiple targeted properties simultaneously but also maintained electrical neutrality in large systems where calculating electrical neutrality is challenging due to combinatorial complexity. As a further potential application, SMOACS should facilitate the exploration of stable structures. Using a compositional formula and various structure candidates they could form, this method minimizes formation energy while maintaining the crystal structure, thus determining the most stable configuration for that formula (see Section A.3). However, the performance of SMOACS heavily depends on the accuracy of property prediction models. Using models based on DFT calculations that underestimate band gaps (see Section A.2) can lead to similar underestimations in the predictions. By adopting more accurate models trained on datasets that are large and developed with more accurate DFT, we may address these challenges.

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

## A APPENDIX

### A.1 DETAILS OF THE SETTINGS IN DENSITY FUNCTIONAL THEORY

We used Density Functional Theory (DFT) to verify the band gaps of materials proposed by SMOACS. We employed the Vienna Ab initio Simulation Package (VASP) (Kresse & Joubert, 1999) version 5.4.4 with Perdew-Burke-Ernzerhof (PBE) exchange-correlation functional (Perdew et al., 1996) and Projector Augmented Wave (PAW) pseudo-potentials (Blöchl, 1994) in all DFT calculations. We used the MPRelaxSet from PyMatGen (Ong et al., 2013) to generate input files: KPOINTS and INCAR.

### A.2 BAND GAP DISCREPANCIES BETWEEN MACHINE LEARNING PREDICTED AND DFT CALCULATED

SMOACS heavily relies on the accuracy of property prediction models. However, we found discrepancies between the values model predicted and the DFT calculated (Fig.A.1). Furthermore, structures relaxed by the MPRelaxSet, which is a parameter set for structural relaxations with VASP provided in PyMatGen, sometimes significantly differ from their proposed forms. There are two possible reasons. First, DFT settings: MEGNet dataset comes from an older version of Materials Project database (Jain et al., 2013). Materials in this database are sometimes updated, and calculation conditions when the MEGNet dataset is created could be different from the current MPRelaxset. We could not reproduce the band gap values in the MEGNet dataset with MPRelaxset. Second, MEG-Net dataset features: All models used in this work are trained on the MEGNet dataset, which is comprised predominantly of stable materials. So, predicting unstable or physically inappropriate structures with these models can lead to inaccurate predictions that may affect the proposed materials. To address these issues, we may need a model trained on a large dataset that includes both stable and unstable structures.

The MEGNet dataset utilizes DFT calculations with PBE functionals that are known to underestimate band gaps. Consequently, when models trained on the MEGNet dataset are used in SMOACS, this tendency may be reflected in the predictions of the proposed materials. This issue may be addressed by constructing a dataset using more accurate band gap calculations, such as HSE06 hybrid functionals (Krukau et al., 2006) and adopting models trained on that dataset. It should be noted that the amount of data available with these accurate calculations is much more limited than for DFT-PBE.

### A.3 A POSSIBLE APPLICATION: IDENTIFYING THE MOST STABLE CRYSTAL STRUCTURE

Our method can optimize energy while specifying the base crystal structure. This property may allow for identifying crystal structures based either solely on the chemical formula or on a combination of the chemical formula and physical properties. This is a Crystal Structure Prediction (CSP) task (Ryan et al., 2018; Miller et al., 2024; Jiao et al., 2024). To verify if this is possible, we experimented to see if the crystal structure of metallic silicon with a zero band gap could be identified. Initially, we extracted structures from the MEGNet dataset that contained only one atom besides Si, using them as the initial structure. The atomic distribution was fixed with a one-hot vector indicating silicon, and only the lattice constants were optimized. The target properties for optimization were a zero band gap and formation energy minimization. We chose silicon structures from the MEGNet dataset with a band gap of 0 eV as the reference and compared these with the optimized structures that exhibited the lowest formation energy. Consequently, we identified structures close to the reference among those optimized for the lowest formation energy.

The results are shown in Table A.1. The reference material of mp-34 is close to optimized candidate No. 2. Similarly, mp-1014212 is close to candidates from No. 4 to No. 12."

### A.4 IMPLEMENTATION DETAILS

To demonstrate that SMOACS can utilize various property prediction models, we selected ALIGNN as a representative of the GNN-based models and Crystalformer as a representative of the

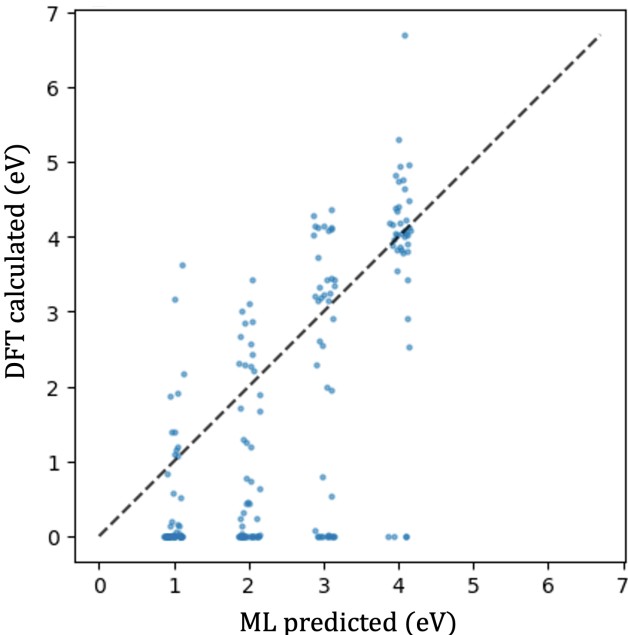

Figure A.1: The discrepancies between band gap values predicted the machine learning model (Crystalformer) and that of DFT calculated.

Table A.1: Reference Si materials (band gap 0 eV) and optimized candidates.

| Materials | $a,b,c$ (Å) | $\alpha,\beta,\gamma$ (°) | predicted formation energy (eV) |
|---|---|---|---|
| (Ref) mp-34 | 2.64, 2.64, 2.47 | 90.0, 90.0, 120 | - |
| (Ref) mp-1014212 | 2.66, 2.66, 2.66 | 109.5, 109.5, 109.5 | - |
| candidate-1 | 2.67, 2.67, 2.94 | 124.0, 124.0, 97.9 | $-0.367$ |
| candidate-2 | 2.50, 2.50, 2.27 | 89.9, 89.9, 134.0 | $-0.359$ |
| candidate-3 | 2.76, 2.76, 2.76 | 115.3, 115.3, 115.3 | $-0.326$ |
| candidate-4 | 2.72, 2.72, 2.72 | 115.0, 115.0, 115.0 | $-0.310$ |
| candidate-5 | 2.71, 2.71, 2.71 | 114.9, 114.9, 114.9 | $-0.310$ |
| candidate-6 | 2.71, 2.71, 2.71 | 114.9, 114.9, 114.9 | $-0.310$ |
| candidate-7 | 2.71, 2.71, 2.71 | 114.9, 114.9, 114.9 | $-0.308$ |
| candidate-8 | 2.73, 2.73, 2.73 | 115.0, 115.0, 115.0 | $-0.308$ |
| candidate-9 | 2.69, 2.69, 2.69 | 114.9, 114.9, 114.9 | $-0.308$ |
| candidate-10 | 2.68, 2.68, 2.68 | 114.8, 114.8, 114.8 | $-0.304$ |
| candidate-11 | 2.64, 2.64, 2.64 | 114.3, 114.3, 114.3 | $-0.266$ |
| candidate-12 | 2.55, 2.55, 2.55 | 113.9, 113.9, 113.9 | $-0.263$ |
| candidate-13 | 2.44, 2.43, 2.44 | 68.8, 64.7, 111.1 | $-0.243$ |
| candidate-14 | 2.42, 2.42, 2.42 | 112.3, 112.3, 112.3 | $-0.239$ |
| candidate-15 | 2.42, 2.42, 2.42 | 112.3, 112.3, 112.3 | $-0.239$ |

Transformer-based models. For both ALIGNN and Crystalformer, we utilized publicly available weights trained on the MEGNet dataset that predict band gaps and formation energies.

The number of optimization steps was 200 for both SMOACS and TPE. The softmax temperature $T$ was linearly decayed from $T = 0.01$ at step 1 to $T = 0.0001$ at step 200. Unless otherwise specified, to prevent extreme crystal structures, the crystal axis lengths $a, b, c$ were clipped to a range of 2 Å to 10 Å, and the angles $\alpha, \beta, \gamma$ were clipped to between 30° and 150°. The types of elements considered ranged from atomic numbers 1 to 98. Unless otherwise noted, the search range

for TPE was aligned with SMOACS, with crystal axis lengths $a, b, c$ ranging from 2 Å to 10 Å and angles $\alpha, \beta, \gamma$ from 30° to 150°. We set the strength parameter $\lambda = 1.0$.

In ALIGNN, bonds are defined using a graph structure. However, because the graph structure is non-differentiable, it cannot be optimized directly. Moreover, as the crystal structure is optimized, the nearest-neighbor atoms may change, potentially rendering the continued use of the same graph structure inappropriate. Therefore, we updated the graph structure multiple times during the optimization process. Considering that the learning rate decay follows a cosine schedule, we updated the graph several times according to a sine schedule, which is the integral of the cosine function.

SMOACS was implemented using PyTorch (Paszke et al., 2019); we used the web-available implementation of FTCP[1] and trained it on the MEGNet dataset. Optuna (Akiba et al., 2019) was used for TPE. We conducted optimizations using ALIGNN and Crystalformer, GNN-based, and transformer-based models, respectively. We used a NVIDIA A100 GPU. We utilized official codes and weights that are available online[2][3].

We trained FTCP from scratch using the MEGNet dataset. We tuned the hyperparameters, including the `max_elms` parameter (the number of types of atoms in the crystal), the `max_sites` parameter (the number of atomic sites in the crystal), and the learning rate. As a result, `max_elms`, `max_sites`, and the learning rate were set to 4, 20, and 0.0001, respectively. Note that the MEGNet dataset contains data with a larger number of element types and sites than these settings, so we did not utilize all 60,000 training samples; however, the reconstruction error score was better with this setting. During inference, after testing several values for the standard deviation of the noise added to the latent variables, we decided to sample from a normal distribution with a mean of 0 and a standard deviation of 0.6.

### A.5 Applicability to Property Prediction Models

This strategy of using atomic distributions discussed in Section 3 is widely applicable to various property prediction models. It readily supports formats such as Crystalformer, where one-hot vectors representing elements are fed into the model. Next, we consider a scenario of using models such as ALIGNN that require atomic representations as input. In this scenario, we treat the inner product of the atomic distribution $\boldsymbol{a}_n$ and the $u$-dimensional representation vector for atoms $\boldsymbol{r}_{\mathrm{atom}}$ ($r_{\mathrm{atom}} \in \mathbb{R}^{K \times u}$) as the atomic representation. In either case, since the output is connected to the learnable atomic distributions through the chain rule of differentiation, we are able to optimize the atomic distribution through backpropagation.

### A.6 Details in Simultaneous Optimization of Targeted Properties regardless of the Crystal Structure

We optimized the band gap regardless of the crystal structure and simultaneously minimized the formation energy. We randomly selected initial crystal structures with up to 10 atomic sites from the MEGNet dataset for SMOACS and TPE, ensuring that each selected structure met the criterion of electrical neutrality. In SMOACS, we selected up to 10 possible oxidation number patterns based on the atom combinations in the initial crystal structure, all of which ensure overall electrical neutrality. the learning rates were set as $\eta_l = 0.01, \eta_C = 0.02, \eta_A = \eta_O = 6.0$ for SMOACS with Crystalformer. For SMOACS with ALIGNN, the learning rates were set as $\eta_l = 0.008, \eta_C = 0.02, \eta_A = \eta_O = 0.0002$. The learning rates were decayed using a cosine annealing schedule.

We updated the graph structure data in ALIGNN 32 times according to a sine schedule, which is the integral of the cosine function. Specifically, we reconstructed the graph structure based on the current crystal configuration at steps [4, 8, 12, 16, 20, 24, 28, 32, 36, 40, 44, 48, 52, 56, 61, 65, 69, 74, 79, 83, 88, 93, 99, 104, 110, 116, 123, 129, 137, 146, 156, 169], as well as during the evaluation after optimization. Then, we constructed masks $\boldsymbol{M}_d$ corresponding to its oxidation pattern $d$. The atomic distribution $\boldsymbol{A}$ and the oxidation state configuration parameter $\boldsymbol{o}$ were initialized with a uniform distribution. The loss functions for the band gap and formation energy in SMOACS use

---

[1]https://github.com/PV-Lab/FTCP

[2]https://github.com/usnistgov/alignn

[3]https://github.com/omron-sinicx/crystalformer

Equation 14. TPE required separate settings for each objective: band gap, formation energy, and electrical neutrality. The objectives for band gap and formation energy were adopted from Equation 13. Additionally, we implemented a binary objective function that assigns a value of 0 if electrical neutrality is achieved and 1 otherwise:

$$L_{\text{neutral}} = \begin{cases} 0 & \text{electrical neutrality} \\ 1 & \text{otherwise} \end{cases} \tag{A.1}$$

TPE used $L_{\text{bg}}$, $L_f$ and $L_{\text{neutral}}$ as objective functions, respectively. FTCP selects initial data from the training dataset where the band gap is close to the target and the formation energy is below -0.5 eV, and then uses an encoder to convert this into latent variables. Next, we add noise to these latent variables using a normal distribution with a mean of 0 and a standard deviation of 0.6, then decode them back into crystal structures for evaluation. Augmented results are shown in Table A.2. We also evaluate the diversity of the proposed materials, as shown in Table A.3.

SMOACS consistently maintained electrical neutrality, provided that extreme geometries causing NaN values during crystal vector calculations did not occur. We calculate the crystal vectors from $a$, $b$, $c$, and $\alpha$, $\beta$, $\gamma$. When the crystal axis lengths or angles are extremely large, computational errors can cause the value under the square root to become a very small negative number, resulting in NaN occurrences. Apart from this, SMOACS consistently maintained electrical neutrality. SMOACS utilizing ALIGNN achieved significantly lower scores compared to when using Crystalformer. We attribute this to the optimization difficulty arising from changes in the shape of the hypersurface of the loss function due to updates to the graph structure. SMOACS demonstrates the ability to generate highly diverse materials.

Table A.2: Experiments on optimizing for various targets of a band gap. The "success rate" is the probability of simultaneously satisfying three conditions: (A) the band gap is optimized within the target range, (B) the formation energy is below -0.5 eV, and (C) the crystal structure is valid. C is achieved when two criteria are met simultaneously: (a) all interatomic distances are greater than 0.5 Å, and (b) the structure is electrically neutral. S(Cry) and S(ALI) denote SMOACS utilizing the Crystalformer and ALIGNN models, respectively. We evaluate each of the proposed materials using all evaluation metrics, and the results are averaged over 256 samples. Higher scores are better across all metrics.

| Target BG (eV) | method | success rate | (A)BG | (B)$E_f$ | (C)STR | (a) neut | (b) 0.5Å |
|---|---|---|---|---|---|---|---|
| 0.50 ±0.04 | S(Cry) | **0.328** | 0.465 | 0.566 | 0.758 | 0.957 | 0.758 |
| | S(ALI) | 0.055 | 0.062 | 0.867 | 0.867 | 0.949 | 0.867 |
| | TPE | 0.004 | 0.945 | 0.059 | 0.066 | 0.070 | 0.910 |
| | FTCP | 0.000 | 0.004 | 1.000 | 0.719 | 0.746 | 0.906 |
| 1.00 ±0.04 | S(Cry) | **0.340** | 0.504 | 0.613 | 0.785 | 0.973 | 0.785 |
| | S(ALI) | 0.047 | 0.059 | 0.848 | 0.805 | 0.926 | 0.805 |
| | TPE | 0.016 | 0.934 | 0.066 | 0.070 | 0.070 | 0.891 |
| | FTCP | 0.004 | 0.004 | 1.000 | 0.699 | 0.730 | 0.891 |
| 1.50 ±0.04 | S(Cry) | **0.387** | 0.543 | 0.672 | 0.824 | 0.980 | 0.824 |
| | S(ALI) | 0.043 | 0.066 | 0.828 | 0.852 | 0.938 | 0.852 |
| | TPE | 0.020 | 0.855 | 0.055 | 0.074 | 0.082 | 0.828 |
| | FTCP | 0.000 | 0.000 | 1.000 | 0.703 | 0.723 | 0.895 |
| 2.00 ±0.04 | S(Cry) | **0.355** | 0.484 | 0.703 | 0.844 | 0.988 | 0.844 |
| | S(ALI) | 0.082 | 0.092 | 0.820 | 0.838 | 0.914 | 0.838 |
| | TPE | 0.020 | 0.789 | 0.062 | 0.086 | 0.086 | 0.812 |
| | FTCP | 0.000 | 0.000 | 1.000 | 0.699 | 0.727 | 0.895 |
| 2.50 ±0.04 | S(Cry) | **0.383** | 0.473 | 0.715 | 0.840 | 0.984 | 0.840 |
| | S(ALI) | 0.051 | 0.059 | 0.809 | 0.793 | 0.898 | 0.793 |
| | TPE | 0.023 | 0.711 | 0.098 | 0.051 | 0.055 | 0.816 |
| | FTCP | 0.004 | 0.004 | 1.000 | 0.695 | 0.707 | 0.902 |
| 3.00 ±0.04 | S(Cry) | **0.301** | 0.375 | 0.699 | 0.828 | 0.992 | 0.828 |
| | S(ALI) | 0.039 | 0.043 | 0.801 | 0.820 | 0.906 | 0.820 |
| | TPE | 0.020 | 0.645 | 0.094 | 0.090 | 0.098 | 0.766 |
| | FTCP | 0.027 | 0.031 | 1.000 | 0.668 | 0.680 | 0.902 |
| 3.50 ±0.04 | S(Cry) | **0.188** | 0.273 | 0.645 | 0.750 | 0.992 | 0.750 |
| | S(ALI) | 0.016 | 0.016 | 0.816 | 0.797 | 0.902 | 0.797 |
| | TPE | 0.012 | 0.586 | 0.059 | 0.055 | 0.059 | 0.730 |
| | FTCP | 0.012 | 0.012 | 1.000 | 0.707 | 0.730 | 0.883 |
| 4.00 ±0.04 | S(Cry) | **0.160** | 0.227 | 0.656 | 0.789 | 1.000 | 0.789 |
| | S(ALI) | 0.023 | 0.023 | 0.805 | 0.797 | 0.902 | 0.797 |
| | TPE | 0.016 | 0.438 | 0.090 | 0.078 | 0.078 | 0.680 |
| | FTCP | 0.035 | 0.043 | 1.000 | 0.676 | 0.691 | 0.895 |

Table A.3: The diversity of the proposed materials. The 'success rate' corresponds to the same 'success rate' as in Table A.2. The 'unique rate' refers to the probability of materials with unique elemental combinations, regardless of the success. The 'unique rate in success' represents the proportion of materials with unique elemental combinations among the successfully optimized materials. The 'unique and novel rate in success' indicates the proportion of materials whose elemental combinations are unique and absent from the MEGNet database among the successfully optimized materials.

| Target BG (eV) | method | success rate | unique rate | unique rate in success | unique and novel rate in success |
|---|---|---|---|---|---|
| 0.50 ±0.04 | S(Cry) | 0.328 | 0.957 | 84/84 | 81/84 |
| | S(ALI) | 0.055 | 0.867 | 14/14 | 13/14 |
| | TPE | 0.004 | 1.000 | 1/1 | 0/1 |
| | FTCP | 0.000 | 0.297 | - | - |
| 1.00 ±0.04 | S(Cry) | 0.340 | 0.973 | 87/87 | 80/87 |
| | S(ALI) | 0.047 | 0.895 | 12/12 | 11/12 |
| | TPE | 0.016 | 1.000 | 4/4 | 1/4 |
| | FTCP | 0.004 | 0.289 | 1/1 | 0/1 |
| 1.50 ±0.04 | S(Cry) | 0.387 | 0.977 | 99/99 | 94/99 |
| | S(ALI) | 0.043 | 0.891 | 11/11 | 11/11 |
| | TPE | 0.020 | 1.000 | 5/5 | 0/5 |
| | FTCP | 0.000 | 0.324 | - | - |
| 2.00 ±0.04 | S(Cry) | 0.355 | 0.984 | 90/91 | 85/91 |
| | S(ALI) | 0.082 | 0.836 | 42/42 | 41/42 |
| | TPE | 0.020 | 1.000 | 5/5 | 1/5 |
| | FTCP | 0.000 | 0.328 | - | - |
| 2.50 ±0.04 | S(Cry) | 0.383 | 0.980 | 98/98 | 90/98 |
| | S(ALI) | 0.051 | 0.793 | 13/13 | 12/13 |
| | TPE | 0.023 | 1.000 | 6/6 | 1/6 |
| | FTCP | 0.004 | 0.328 | 1/1 | 0/1 |
| 3.00 ±0.04 | S(Cry) | 0.301 | 0.992 | 77/77 | 72/77 |
| | S(ALI) | 0.039 | 0.770 | 9/10 | 6/10 |
| | TPE | 0.020 | 0.992 | 4/5 | 2/5 |
| | FTCP | 0.027 | 0.359 | 6/7 | 0/7 |
| 3.50 ±0.04 | S(Cry) | 0.188 | 0.992 | 48/48 | 47/48 |
| | S(ALI) | 0.016 | 0.793 | 4/4 | 4/4 |
| | TPE | 0.012 | 1.000 | 3/3 | 1/3 |
| | FTCP | 0.012 | 0.309 | 3/3 | 0/3 |
| 4.00 ±0.04 | S(Cry) | 0.160 | 0.996 | 41/41 | 38/41 |
| | S(ALI) | 0.023 | 0.797 | 6/6 | 6/6 |
| | TPE | 0.016 | 0.996 | 4/4 | 1/4 |
| | FTCP | 0.035 | 0.348 | 7/9 | 0/9 |

### A.7 DETAILS IN SIMULTANEOUS OPTIMIZATION OF TARGETED PROPERTIES WHILE PRESERVING PEROVSKITE STRUCTURES

As discussed in Section 3.3, due to the arbitrariness in the numerical values of the lattice constant and coordinates of perovskite structures, we evaluated whether the optimized structures approximated typical perovskite configurations. First of all, fractional coordinates typical of perovskite structures are as follows: $(0.5, 0.5, 0.5)$ at the A site, $(0.0, 0.0, 0.0)$ at the B site, and $(0.5, 0.0, 0.0), (0.0, 0.5, 0.0), (0.0, 0.0, 0.5)$ at the three X sites. We established criteria for the optimized x, y, and z coordinates to be within a deviation $\epsilon$ from these standard values. The perovskite structure $CaCu_3Ti_4O_{12}$ exhibits a slightly distorted configuration, with the x-coordinate of the oxygen atoms deviating by approximately 10% from their typical positions (Božin et al., 2004). To explore new structures, we set $\epsilon = 0.15$, allowing for a slightly greater distortion. We considered the optimized coordinates successful if the x, y, and z coordinates of each site fell within $\pm\epsilon$. Additionally, the angles between the crystal axes of typical perovskite structures are close to 90°. Therefore, angles between 85° and 95° were established as a criterion.

Using $t$ values from typical perovskite structures ($BaCeO_3$:0.857, $SrTiO_3$: 0.910 and $BaTiO_3$: 0.970), we established a tolerance factor range of $0.8 \leq t \leq 1.0$ as the criterion for success. The ionic radius of the X site was calculated as the average of the radii of the three X sites. We took the values for the ionic radii from PyMatGen (Ong et al., 2013).

In this experiments, the learning rates were set as $\eta_l = 0.01, \eta_C = 0.02, \eta_A = \eta_O = 6.0$ for SMOACS with Crystalformer. For SMOACS with ALIGNN, the learning rates were set as $\eta_l = 0.5, \eta_C = 0.002, \eta_A = \eta_O = 0.00008$. We reconstructed the graph structure 46 times during the optimizations. The learning rates were decayed using a cosine annealing schedule.

Due to the limited number of perovskite structure data points in the MEGNet dataset, we generated 256 random perovskite structures as initial values for SMOACS. These structures have crystal axis angles $\alpha$, $\beta$, and $\gamma$ at 90° and axis lengths $a$, $b$, and $c$ randomly generated between 2 Å and 10 Å. Their initial fractional coordinates correspond to those typical of perovskite structures: $(0.5, 0.5, 0.5)$ for the A site, $(0.0, 0.0, 0.0)$ for the B site, and $(0.5, 0.0, 0.0)$, $(0.0, 0.5, 0.0)$, and $(0.0, 0.0, 0.5)$ for the three X sites. Similarly, TPE optimized perovskite structures by setting the crystal axis angles at 90° and optimizing the axis lengths $a, b, c$ between 2 Å and 10 Å. We also limited element species for each site in TPE. Specifically, the elements are restricted by oxidation numbers: +1 and +2 for site A, +2 and +4 for site B, and −1 and −2 for site X. For FTCP, we initially selected data points where the crystal axis angles were at 90°, and all sites conformed to the typical fractional coordinates of perovskite structures; these were then converted into latent variables. Subsequently, we applied noise using a normal distribution with a mean of 0 and a standard deviation of 0.6 to the latent variables. Finally, we decoded the latent variables back into crystal structures for evaluation.

SMOACS conducted optimization using Eq 17. For $t$, TPE used another objective function:

$$L_t^{\text{TPE}} = \begin{cases} 0 & (0.8 \leq t \leq 1.0) \\ 1 & \text{otherwise} \end{cases} \tag{A.2}$$

TPE used $L_{\text{bg}}$, $L_f$, $L_{\text{neutral}}$ and $L_t^{\text{TPE}}$ as objective functions, respectively. Augmented results are shown in Table A.4. We also evaluate the diversity of the proposed materials, as shown in Table A.5.

Table A.4: Experiments on optimizing various band gaps while preserving perovskite structures. The "success rate" reflects the probability of simultaneously satisfying four criteria: (A) the band gap is optimized within the target range, (B) the formation energy is below -0.5 eV, (C) the crystal structure is valid, and (D) approximating a valid perovskite structure. Criteria (A), (B), and (C) are consistent with those outlined in Table 2. The (D) is achieved when three criteria are met simultaneously: (c) the tolerance factor $t$ is between 0.8 and 1.0, (d) coordinates are within $\pm 0.15$ of typical perovskite structure coordinates, and (e) axis angles are from 85° to 95°. We evaluate each of the proposed materials using all evaluation metrics, and the results are averaged over 256 samples. Higher scores are better across all metrics.

| Target BG (eV) | method | success rate | (A)BG | (B)$E_f$ | (C)STR | (a) neut | (b) 0.5Å | (D)PS | (c) tole | (d) angles | (e) coord |
|---|---|---|---|---|---|---|---|---|---|---|---|
| 0.50 ±0.04 | S(Cry) | **0.113** | 0.477 | 0.410 | 0.965 | 1.000 | 0.965 | 0.500 | 0.500 | 1.000 | 1.000 |
| | S(ALI) | 0.090 | 0.211 | 0.535 | 1.000 | 1.000 | 1.000 | 0.500 | 0.500 | 1.000 | 1.000 |
| | TPE | 0.027 | 1.000 | 0.137 | 0.535 | 0.535 | 1.000 | 0.648 | 0.648 | 1.000 | 1.000 |
| | FTCP | 0.004 | 0.023 | 1.000 | 0.836 | 0.840 | 0.938 | 0.258 | 0.508 | 0.441 | 0.285 |
| 1.00 ±0.04 | S(Cry) | **0.152** | 0.457 | 0.422 | 0.961 | 1.000 | 0.961 | 0.559 | 0.559 | 1.000 | 1.000 |
| | S(ALI) | 0.062 | 0.168 | 0.484 | 1.000 | 1.000 | 1.000 | 0.531 | 0.531 | 1.000 | 1.000 |
| | TPE | 0.012 | 1.000 | 0.137 | 0.395 | 0.395 | 1.000 | 0.664 | 0.664 | 1.000 | 1.000 |
| | FTCP | 0.004 | 0.023 | 1.000 | 0.859 | 0.863 | 0.965 | 0.215 | 0.531 | 0.418 | 0.270 |
| 1.50 ±0.04 | S(Cry) | **0.148** | 0.422 | 0.461 | 0.984 | 1.000 | 0.984 | 0.578 | 0.578 | 1.000 | 1.000 |
| | S(ALI) | 0.070 | 0.219 | 0.652 | 1.000 | 1.000 | 1.000 | 0.629 | 0.629 | 1.000 | 1.000 |
| | TPE | 0.023 | 0.992 | 0.281 | 0.293 | 0.293 | 1.000 | 0.523 | 0.523 | 1.000 | 1.000 |
| | FTCP | 0.000 | 0.016 | 1.000 | 0.895 | 0.906 | 0.965 | 0.242 | 0.547 | 0.418 | 0.320 |
| 2.00 ±0.04 | S(Cry) | **0.188** | 0.426 | 0.516 | 0.988 | 1.000 | 0.988 | 0.613 | 0.613 | 1.000 | 1.000 |
| | S(ALI) | 0.090 | 0.188 | 0.980 | 1.000 | 1.000 | 1.000 | 0.625 | 0.625 | 1.000 | 1.000 |
| | TPE | 0.027 | 0.977 | 0.266 | 0.266 | 0.266 | 1.000 | 0.547 | 0.547 | 1.000 | 1.000 |
| | FTCP | 0.004 | 0.004 | 1.000 | 0.891 | 0.898 | 0.980 | 0.281 | 0.551 | 0.473 | 0.324 |
| 2.50 ±0.04 | S(Cry) | **0.152** | 0.285 | 0.516 | 0.988 | 1.000 | 0.988 | 0.625 | 0.625 | 1.000 | 1.000 |
| | S(ALI) | 0.113 | 0.184 | 0.938 | 1.000 | 1.000 | 1.000 | 0.625 | 0.625 | 1.000 | 1.000 |
| | TPE | 0.016 | 0.918 | 0.281 | 0.352 | 0.352 | 1.000 | 0.387 | 0.387 | 1.000 | 1.000 |
| | FTCP | 0.008 | 0.012 | 0.996 | 0.879 | 0.898 | 0.953 | 0.250 | 0.543 | 0.441 | 0.289 |
| 3.00 ±0.04 | S(Cry) | 0.102 | 0.219 | 0.508 | 0.992 | 1.000 | 0.992 | 0.621 | 0.621 | 1.000 | 1.000 |
| | S(ALI) | **0.141** | 0.273 | 0.938 | 1.000 | 1.000 | 1.000 | 0.625 | 0.625 | 1.000 | 1.000 |
| | TPE | 0.035 | 0.875 | 0.293 | 0.234 | 0.234 | 1.000 | 0.316 | 0.316 | 1.000 | 1.000 |
| | FTCP | 0.008 | 0.008 | 1.000 | 0.898 | 0.906 | 0.969 | 0.246 | 0.543 | 0.418 | 0.316 |
| 3.50 ±0.04 | S(Cry) | 0.070 | 0.145 | 0.516 | 0.996 | 1.000 | 0.996 | 0.629 | 0.629 | 1.000 | 1.000 |
| | S(ALI) | **0.176** | 0.195 | 0.961 | 1.000 | 1.000 | 1.000 | 0.668 | 0.668 | 1.000 | 1.000 |
| | TPE | 0.016 | 0.711 | 0.266 | 0.184 | 0.184 | 1.000 | 0.285 | 0.285 | 1.000 | 1.000 |
| | FTCP | 0.004 | 0.008 | 1.000 | 0.895 | 0.910 | 0.953 | 0.238 | 0.531 | 0.441 | 0.277 |
| 4.00 ±0.04 | S(Cry) | 0.051 | 0.094 | 0.512 | 0.992 | 1.000 | 0.992 | 0.625 | 0.625 | 1.000 | 1.000 |
| | S(ALI) | **0.180** | 0.285 | 0.961 | 1.000 | 1.000 | 1.000 | 0.605 | 0.605 | 1.000 | 1.000 |
| | TPE | 0.020 | 0.539 | 0.336 | 0.215 | 0.215 | 1.000 | 0.227 | 0.227 | 1.000 | 1.000 |
| | FTCP | 0.000 | 0.004 | 0.996 | 0.883 | 0.887 | 0.949 | 0.238 | 0.535 | 0.410 | 0.305 |

Table A.5: The diversity of the proposed materials. The 'success rate' corresponds to the same 'success rate' as in Table A.4. The 'unique rate' refers to the probability of materials with unique elemental combinations, regardless of the success. The 'unique rate in success' represents the proportion of materials with unique elemental combinations among the successfully optimized materials. The 'unique and novel rate in success' indicates the proportion of materials whose elemental combinations are unique and absent from the MEGNet database among the successfully optimized materials.

| Target BG (eV) | method | success rate | unique rate | unique rate in success | unique and novel rate in success |
|---|---|---|---|---|---|
| 0.50 ±0.04 | S(Cry) | 0.113 | 0.898 | 28/29 | 27/29 |
| | S(ALI) | 0.090 | 0.398 | 14/23 | 13/23 |
| | TPE | 0.027 | 0.984 | 7/7 | 7/7 |
| | FTCP | 0.004 | 0.352 | 1/1 | 0/1 |
| 1.00 ±0.04 | S(Cry) | 0.152 | 0.891 | 38/39 | 37/39 |
| | S(ALI) | 0.062 | 0.562 | 14/16 | 12/16 |
| | TPE | 0.012 | 0.996 | 3/3 | 3/3 |
| | FTCP | 0.004 | 0.336 | 1/1 | 0/1 |
| 1.50 ±0.04 | S(Cry) | 0.148 | 0.891 | 38/38 | 38/38 |
| | S(ALI) | 0.070 | 0.555 | 16/18 | 15/18 |
| | TPE | 0.023 | 1.000 | 6/6 | 6/6 |
| | FTCP | 0.000 | 0.285 | - | - |
| 2.00 ±0.04 | S(Cry) | 0.188 | 0.902 | 47/48 | 46/48 |
| | S(ALI) | 0.090 | 0.445 | 17/23 | 17/23 |
| | TPE | 0.027 | 0.996 | 7/7 | 7/7 |
| | FTCP | 0.004 | 0.309 | 1/1 | 0/1 |
| 2.50 ±0.04 | S(Cry) | 0.152 | 0.898 | 38/39 | 36/39 |
| | S(ALI) | 0.113 | 0.598 | 25/29 | 23/29 |
| | TPE | 0.016 | 0.992 | 4/4 | 4/4 |
| | FTCP | 0.008 | 0.309 | 2/2 | 0/2 |
| 3.00 ±0.04 | S(Cry) | 0.102 | 0.902 | 26/26 | 26/26 |
| | S(ALI) | 0.141 | 0.465 | 24/36 | 21/36 |
| | TPE | 0.035 | 0.992 | 9/9 | 9/9 |
| | FTCP | 0.008 | 0.285 | 2/2 | 0/2 |
| 3.50 ±0.04 | S(Cry) | 0.070 | 0.902 | 18/18 | 17/18 |
| | S(ALI) | 0.176 | 0.387 | 14/45 | 12/45 |
| | TPE | 0.016 | 0.988 | 4/4 | 4/4 |
| | FTCP | 0.004 | 0.301 | 1/1 | 0/1 |
| 4.00 ±0.04 | S(Cry) | 0.051 | 0.902 | 13/13 | 13/13 |
| | S(ALI) | 0.180 | 0.387 | 20/46 | 14/46 |
| | TPE | 0.020 | 0.977 | 5/5 | 5/5 |
| | FTCP | 0.000 | 0.312 | - | - |

### A.8  DETAILS IN OPTIMIZING LARGE ATOMIC CONFIGURATIONS

We conducted experiments on $3 \times 3 \times 3$ perovskite structures containing 135 atom sites, expanded from a unit cell with five atom sites. As the cell size increased, the range for the crystal lattice dimensions $a, b, c$ in SMOACS and TPE was set from 6 Å to 30 Å for the $3 \times 3 \times 3$ structure. Similarly, the range of coordinate variations $\epsilon$ was set to 0.05. Aside from these changes, the experimental conditions remained consistent with those described in Section 4.3. In this experiments, the learning rates were set as $\eta_l = 0.003, \eta_C = 0.005, \eta_A = \eta_O = 2.0$ for SMOACS with Crystalformer. For SMOACS with ALIGNN, the learning rates were set as $\eta_l = 5.000, \eta_C = 0.002, \eta_A = \eta_O = 0.00005$. We reconstructed the graph structure 41 times during the optimizations. The learning rates were decayed using a cosine annealing schedule. Augmented results are shown in Table A.6.

Table A.6: Experiments optimizing for various band gaps while preserving a $3 \times 3 \times 3$ perovskite structure. We included only TPE, which showed better performance in Section 4.3, for comparison. Evaluation methods are based on those described in Table 3.

| Target BG (eV) | method | success rate | (A)BG | (B)E$_f$ | (C)STR | (a) neut | (b) 0.5Å | (D)PS | (c) tole | (d) angles | (e) coord |
|---|---|---|---|---|---|---|---|---|---|---|---|
| 0.50 ±0.04 | S(Cry) | 0.156 | 0.734 | 0.547 | 0.968 | 1.00 | 0.969 | 0.570 | 0.570 | 1.000 | 1.000 |
| | S(ALI) | **0.188** | 0.234 | 0.812 | 0.687 | 1.00 | 0.688 | 0.789 | 0.789 | 1.000 | 1.000 |
| | TPE(/N) | 0.000 | 1.000 | 0.000 | - | N/A | 1.000 | 0.609 | 0.609 | 1.000 | 1.000 |
| 1.00 ±0.04 | S(Cry) | 0.070 | 0.250 | 0.469 | 0.945 | 1.00 | 0.945 | 0.586 | 0.586 | 1.000 | 1.000 |
| | S(ALI) | **0.094** | 0.133 | 0.828 | 0.703 | 1.00 | 0.703 | 0.625 | 0.625 | 1.000 | 1.000 |
| | TPE(/N) | 0.000 | 0.125 | 0.000 | - | N/A | 0.992 | 0.242 | 0.242 | 1.000 | 1.000 |
| 1.50 ±0.04 | S(Cry) | 0.047 | 0.125 | 0.422 | 0.953 | 1.00 | 0.953 | 0.617 | 0.617 | 1.000 | 1.000 |
| | S(ALI) | **0.062** | 0.086 | 0.867 | 0.726 | 1.00 | 0.727 | 0.586 | 0.586 | 1.000 | 1.000 |
| | TPE(/N) | 0.000 | 0.141 | 0.000 | - | N/A | 1.000 | 0.180 | 0.180 | 1.000 | 1.000 |
| 2.00 ±0.04 | S(Cry) | 0.023 | 0.055 | 0.406 | 0.976 | 1.00 | 0.977 | 0.633 | 0.633 | 1.000 | 1.000 |
| | S(ALI) | **0.055** | 0.102 | 1.000 | 0.710 | 1.00 | 0.711 | 0.594 | 0.594 | 1.000 | 1.000 |
| | TPE(/N) | 0.000 | 0.125 | 0.000 | - | N/A | 0.984 | 0.242 | 0.242 | 1.000 | 1.000 |
| 2.50 ±0.04 | S(Cry) | 0.023 | 0.039 | 0.438 | 0.984 | 1.00 | 0.984 | 0.664 | 0.664 | 1.000 | 1.000 |
| | S(ALI) | **0.102** | 0.172 | 1.000 | 0.703 | 1.00 | 0.703 | 0.812 | 0.812 | 1.000 | 1.000 |
| | TPE(/N) | 0.000 | 0.023 | 0.000 | - | N/A | 0.984 | 0.156 | 0.156 | 1.000 | 1.000 |
| 3.00 ±0.04 | S(Cry) | 0.016 | 0.047 | 0.602 | 1.00 | 1.00 | 1.000 | 0.383 | 0.383 | 1.000 | 1.000 |
| | S(ALI) | **0.125** | 0.188 | 0.992 | 0.726 | 1.00 | 0.727 | 0.664 | 0.664 | 1.000 | 1.000 |
| | TPE(/N) | 0.000 | 0.023 | 0.000 | - | N/A | 0.984 | 0.273 | 0.273 | 1.000 | 1.000 |
| 3.50 ±0.04 | S(Cry) | 0.008 | 0.008 | 0.445 | 0.984 | 1.00 | 0.984 | 0.672 | 0.672 | 1.000 | 1.000 |
| | S(ALI) | **0.156** | 0.250 | 1.000 | 0.75 | 1.00 | 0.750 | 0.734 | 0.734 | 1.000 | 1.000 |
| | TPE(/N) | 0.000 | 0.000 | 0.000 | - | N/A | 0.992 | 0.195 | 0.195 | 1.000 | 1.000 |
| 4.00 ±0.04 | S(Cry) | 0.008 | 0.008 | 0.445 | 0.984 | 1.00 | 0.984 | 0.672 | 0.672 | 1.000 | 1.000 |
| | S(ALI) | **0.219** | 0.305 | 1.000 | 0.773 | 1.00 | 0.773 | 0.852 | 0.852 | 1.000 | 1.000 |
| | TPE(/N) | 0.000 | 0.000 | 0.000 | - | N/A | 0.969 | 0.180 | 0.180 | 1.000 | 1.000 |

## A.9 ELECTRICAL NEUTRALITY

In assessing electrical neutrality, a compound was considered neutral if the sum of the oxidation numbers for the atoms at each site equaled zero. For example, $Fe_3O_4$ is electrically neutral because the configuration $[Fe, Fe, Fe, O, O, O, O]$ can assume oxidation numbers of $[+2, +3, +3, -2, -2, -2, -2]$ that sum to zero. Previous study (Xie et al., 2022) employed SMACT(Davies et al., 2019) to assess electrical neutrality; however, SMACT includes some oxidation numbers, like the +7 state of chlorine, which are extremely rare and potentially less reliable. We restricted our analysis to commonly occurring oxidation numbers, selecting those found at the intersection of SMACT and PyMatGen. A list of the elements and their corresponding oxidation numbers employed in this study is shown in Table A.7, Table A.8, and Table A.9. In these tables, the 'SMACT' indicates oxidation numbers from `smact.Element`. The 'icsd' and 'common' indicate oxidation numbers from `icsd_oxidation_state` and `common_oxidation_states` in `pymatgen.core.periodic_table.Element`, respectively. 'Ours' represents the oxidation numbers we used in this paper.

Table A.7: The List of oxidation numbers from Hydrogen (H) to Krypton (Kr).

| Z | Elm | SMACT | icsd | common | Ours |
|---|---|---|---|---|---|
| 1 | H | $\{-1, 1\}$ | $\{-1, 1\}$ | $\{-1, 1\}$ | $\{-1, 1\}$ |
| 2 | He | $\{\}$ | $\{\}$ | $\{\}$ | $\{\}$ |
| 3 | Li | $\{1\}$ | $\{1\}$ | $\{1\}$ | $\{1\}$ |
| 4 | Be | $\{1, 2\}$ | $\{2\}$ | $\{2\}$ | $\{2\}$ |
| 5 | B | $\{1, 2, 3\}$ | $\{-3, 3\}$ | $\{3\}$ | $\{3\}$ |
| 6 | C | $\{-4, -3, -2, -1, 1, 2, 3, 4\}$ | $\{-4, -3, -2, 2, 3, 4\}$ | $\{-4, 4\}$ | $\{-4, 4\}$ |
| 7 | N | $\{-3, -2, -1, 1, 2, 3, 4, 5\}$ | $\{-3, -2, -1, 1, 3, 5\}$ | $\{-3, 3, 5\}$ | $\{-3, 3, 5\}$ |
| 8 | O | $\{-2, -1, 1, 2\}$ | $\{-2\}$ | $\{-2\}$ | $\{-2\}$ |
| 9 | F | $\{-1\}$ | $\{-1\}$ | $\{-1\}$ | $\{-1\}$ |
| 10 | Ne | $\{\}$ | $\{\}$ | $\{\}$ | $\{\}$ |
| 11 | Na | $\{-1, 1\}$ | $\{1\}$ | $\{1\}$ | $\{1\}$ |
| 12 | Mg | $\{1, 2\}$ | $\{2\}$ | $\{2\}$ | $\{2\}$ |
| 13 | Al | $\{1, 2, 3\}$ | $\{3\}$ | $\{3\}$ | $\{3\}$ |
| 14 | Si | $\{-4, -3, -2, -1, 1, 2, 3, 4\}$ | $\{-4, 4\}$ | $\{-4, 4\}$ | $\{-4, 4\}$ |
| 15 | P | $\{-3, -2, -1, 1, 2, 3, 4, 5\}$ | $\{-3, -2, -1, 3, 4, 5\}$ | $\{-3, 3, 5\}$ | $\{-3, 3, 5\}$ |
| 16 | S | $\{-2, -1, 1, 2, 3, 4, 5, 6\}$ | $\{-2, -1, 2, 4, 6\}$ | $\{-2, 2, 4, 6\}$ | $\{-2, 2, 4, 6\}$ |
| 17 | Cl | $\{-1, 1, 2, 3, 4, 5, 6, 7\}$ | $\{-1\}$ | $\{-1, 1, 3, 5, 7\}$ | $\{-1\}$ |
| 18 | Ar | $\{\}$ | $\{\}$ | $\{\}$ | $\{\}$ |
| 19 | K | $\{-1, 1\}$ | $\{1\}$ | $\{1\}$ | $\{1\}$ |
| 20 | Ca | $\{1, 2\}$ | $\{2\}$ | $\{2\}$ | $\{2\}$ |
| 21 | Sc | $\{1, 2, 3\}$ | $\{2, 3\}$ | $\{3\}$ | $\{3\}$ |
| 22 | Ti | $\{-1, 1, 2, 3, 4\}$ | $\{2, 3, 4\}$ | $\{4\}$ | $\{4\}$ |
| 23 | V | $\{-1, 1, 2, 3, 4, 5\}$ | $\{2, 3, 4, 5\}$ | $\{5\}$ | $\{5\}$ |
| 24 | Cr | $\{-2, -1, 1, 2, 3, 4, 5, 6\}$ | $\{2, 3, 4, 5, 6\}$ | $\{3, 6\}$ | $\{3, 6\}$ |
| 25 | Mn | $\{-3, -2, -1, 1, 2, 3, 4, 5, 6, 7\}$ | $\{2, 3, 4, 7\}$ | $\{2, 4, 7\}$ | $\{2, 4, 7\}$ |
| 26 | Fe | $\{-2, -1, 1, 2, 3, 4, 5, 6\}$ | $\{2, 3\}$ | $\{2, 3\}$ | $\{2, 3\}$ |
| 27 | Co | $\{-1, 1, 2, 3, 4, 5\}$ | $\{1, 2, 3, 4\}$ | $\{2, 3\}$ | $\{2, 3\}$ |
| 28 | Ni | $\{-1, 1, 2, 3, 4\}$ | $\{1, 2, 3, 4\}$ | $\{2\}$ | $\{2\}$ |
| 29 | Cu | $\{1, 2, 3, 4\}$ | $\{1, 2, 3\}$ | $\{2\}$ | $\{2\}$ |
| 30 | Zn | $\{1, 2\}$ | $\{2\}$ | $\{2\}$ | $\{2\}$ |
| 31 | Ga | $\{1, 2, 3\}$ | $\{2, 3\}$ | $\{3\}$ | $\{3\}$ |
| 32 | Ge | $\{-4, -3, -2, -1, 1, 2, 3, 4\}$ | $\{2, 3, 4\}$ | $\{-4, 2, 4\}$ | $\{2, 4\}$ |
| 33 | As | $\{-3, 1, 2, 3, 5\}$ | $\{-3, -2, -1, 2, 3, 5\}$ | $\{-3, 3, 5\}$ | $\{-3, 3, 5\}$ |
| 34 | Se | $\{-2, 1, 2, 4, 6\}$ | $\{-2, -1, 4, 6\}$ | $\{-2, 2, 4, 6\}$ | $\{-2, 4, 6\}$ |
| 35 | Br | $\{-1, 1, 2, 3, 4, 5, 7\}$ | $\{-1, 5\}$ | $\{-1, 1, 3, 5, 7\}$ | $\{-1, 5\}$ |
| 36 | Kr | $\{2\}$ | $\{\}$ | $\{\}$ | $\{\}$ |

Table A.8: The List of oxidation numbers from Rubidium (Rb) to Radon (Rn).

| Z | Elm | SMACT | icsd | common | Ours |
|---|-----|-------|------|--------|------|
| 37 | Rb | $\{-1, 1\}$ | $\{1\}$ | $\{1\}$ | $\{1\}$ |
| 38 | Sr | $\{1, 2\}$ | $\{2\}$ | $\{2\}$ | $\{2\}$ |
| 39 | Y | $\{1, 2, 3\}$ | $\{3\}$ | $\{3\}$ | $\{3\}$ |
| 40 | Zr | $\{1, 2, 3, 4\}$ | $\{2, 3, 4\}$ | $\{4\}$ | $\{4\}$ |
| 41 | Nb | $\{-1, 1, 2, 3, 4, 5\}$ | $\{2, 3, 4, 5\}$ | $\{5\}$ | $\{5\}$ |
| 42 | Mo | $\{-2, -1, 1, 2, 3, 4, 5, 6\}$ | $\{2, 3, 4, 5, 6\}$ | $\{4, 6\}$ | $\{4, 6\}$ |
| 43 | Tc | $\{-3, -1, 1, 2, 3, 4, 5, 6, 7\}$ | $\{\}$ | $\{4, 7\}$ | $\{\}$ |
| 44 | Ru | $\{-2, 1, 2, 3, 4, 5, 6, 7, 8\}$ | $\{2, 3, 4, 5, 6\}$ | $\{3, 4\}$ | $\{3, 4\}$ |
| 45 | Rh | $\{-1, 1, 2, 3, 4, 5, 6\}$ | $\{3, 4\}$ | $\{3\}$ | $\{3\}$ |
| 46 | Pd | $\{1, 2, 4, 6\}$ | $\{2, 4\}$ | $\{2, 4\}$ | $\{2, 4\}$ |
| 47 | Ag | $\{1, 2, 3, 4\}$ | $\{1, 2, 3\}$ | $\{1\}$ | $\{1\}$ |
| 48 | Cd | $\{1, 2\}$ | $\{2\}$ | $\{2\}$ | $\{2\}$ |
| 49 | In | $\{1, 2, 3\}$ | $\{1, 2, 3\}$ | $\{3\}$ | $\{3\}$ |
| 50 | Sn | $\{-4, 2, 4\}$ | $\{2, 3, 4\}$ | $\{-4, 2, 4\}$ | $\{2, 4\}$ |
| 51 | Sb | $\{-3, 3, 5\}$ | $\{-3, -2, -1, 3, 5\}$ | $\{-3, 3, 5\}$ | $\{-3, 3, 5\}$ |
| 52 | Te | $\{-2, 2, 4, 5, 6\}$ | $\{-2, -1, 4, 6\}$ | $\{-2, 2, 4, 6\}$ | $\{-2, 4, 6\}$ |
| 53 | I | $\{-1, 1, 3, 4, 5, 7\}$ | $\{-1, 5\}$ | $\{-1, 1, 3, 5, 7\}$ | $\{-1, 5\}$ |
| 54 | Xe | $\{1, 2, 4, 6, 8\}$ | $\{\}$ | $\{\}$ | $\{\}$ |
| 55 | Cs | $\{-1, 1\}$ | $\{1\}$ | $\{1\}$ | $\{1\}$ |
| 56 | Ba | $\{2\}$ | $\{2\}$ | $\{2\}$ | $\{2\}$ |
| 57 | La | $\{2, 3\}$ | $\{2, 3\}$ | $\{3\}$ | $\{3\}$ |
| 58 | Ce | $\{2, 3, 4\}$ | $\{3, 4\}$ | $\{3, 4\}$ | $\{3, 4\}$ |
| 59 | Pr | $\{2, 3, 4\}$ | $\{3, 4\}$ | $\{3\}$ | $\{3\}$ |
| 60 | Nd | $\{2, 3, 4\}$ | $\{2, 3\}$ | $\{3\}$ | $\{3\}$ |
| 61 | Pm | $\{2, 3\}$ | $\{\}$ | $\{3\}$ | $\{\}$ |
| 62 | Sm | $\{2, 3\}$ | $\{2, 3\}$ | $\{3\}$ | $\{3\}$ |
| 63 | Eu | $\{2, 3\}$ | $\{2, 3\}$ | $\{2, 3\}$ | $\{2, 3\}$ |
| 64 | Gd | $\{1, 2, 3\}$ | $\{3\}$ | $\{3\}$ | $\{3\}$ |
| 65 | Tb | $\{1, 2, 3, 4\}$ | $\{3, 4\}$ | $\{3\}$ | $\{3\}$ |
| 66 | Dy | $\{2, 3, 4\}$ | $\{3\}$ | $\{3\}$ | $\{3\}$ |
| 67 | Ho | $\{2, 3\}$ | $\{3\}$ | $\{3\}$ | $\{3\}$ |
| 68 | Er | $\{2, 3\}$ | $\{3\}$ | $\{3\}$ | $\{3\}$ |
| 69 | Tm | $\{2, 3\}$ | $\{3\}$ | $\{3\}$ | $\{3\}$ |
| 70 | Yb | $\{2, 3\}$ | $\{2, 3\}$ | $\{3\}$ | $\{3\}$ |
| 71 | Lu | $\{3\}$ | $\{3\}$ | $\{3\}$ | $\{3\}$ |
| 72 | Hf | $\{2, 3, 4\}$ | $\{4\}$ | $\{4\}$ | $\{4\}$ |
| 73 | Ta | $\{-1, 2, 3, 4, 5\}$ | $\{3, 4, 5\}$ | $\{5\}$ | $\{5\}$ |
| 74 | W | $\{-2, -1, 1, 2, 3, 4, 5, 6\}$ | $\{2, 3, 4, 5, 6\}$ | $\{4, 6\}$ | $\{4, 6\}$ |
| 75 | Re | $\{-3, -1, 1, 2, 3, 4, 5, 6, 7\}$ | $\{3, 4, 5, 6, 7\}$ | $\{4\}$ | $\{4\}$ |
| 76 | Os | $\{-2, -1, 1, 2, 3, 4, 5, 6, 7, 8\}$ | $\{\}$ | $\{4\}$ | $\{\}$ |
| 77 | Ir | $\{-3, -1, 1, 2, 3, 4, 5, 6, 7, 8\}$ | $\{3, 4, 5\}$ | $\{3, 4\}$ | $\{3, 4\}$ |
| 78 | Pt | $\{-2, -1, 1, 2, 3, 4, 5, 6\}$ | $\{\}$ | $\{2, 4\}$ | $\{\}$ |
| 79 | Au | $\{-1, 1, 2, 3, 5\}$ | $\{\}$ | $\{3\}$ | $\{\}$ |
| 80 | Hg | $\{1, 2, 4\}$ | $\{1, 2\}$ | $\{1, 2\}$ | $\{1, 2\}$ |
| 81 | Tl | $\{-1, 1, 3\}$ | $\{1, 3\}$ | $\{1, 3\}$ | $\{1, 3\}$ |
| 82 | Pb | $\{-4, 2, 4\}$ | $\{2, 4\}$ | $\{2, 4\}$ | $\{2, 4\}$ |
| 83 | Bi | $\{-3, 1, 3, 5, 7\}$ | $\{1, 2, 3, 5\}$ | $\{3\}$ | $\{3\}$ |
| 84 | Po | $\{-2, 2, 4, 5, 6\}$ | $\{\}$ | $\{-2, 2, 4\}$ | $\{\}$ |
| 85 | At | $\{-1, 1, 3, 5, 7\}$ | $\{\}$ | $\{-1, 1\}$ | $\{\}$ |
| 86 | Rn | $\{2, 6\}$ | $\{\}$ | $\{\}$ | $\{\}$ |

Table A.9: The List of oxidation numbers from Francium (Fr) to Californium (Cf)

| Z | Elm | smact | icsd | common | Ours |
|---|---|---|---|---|---|
| 87 | Fr | {1} | {} | {1} | {} |
| 88 | Ra | {2} | {} | {2} | {} |
| 89 | Ac | {2, 3} | {} | {3} | {} |
| 90 | Th | {2, 3, 4} | {4} | {4} | {4} |
| 91 | Pa | {2, 3, 4, 5} | {} | {5} | {} |
| 92 | U | {2, 3, 4, 5, 6} | {3, 4, 5, 6} | {6} | {6} |
| 93 | Np | {3, 4, 5, 6, 7} | {} | {5} | {} |
| 94 | Pu | {2, 3, 4, 5, 6, 7, 8} | {} | {4} | {} |
| 95 | Am | {2, 3, 4, 5, 6, 7} | {} | {3} | {} |
| 96 | Cm | {2, 3, 4, 6, 8} | {} | {3} | {} |
| 97 | Bk | {2, 3, 4} | {} | {3} | {} |
| 98 | Cf | {2, 3, 4} | {} | {3} | {} |

## A.10 ACCURACTE BAND GAP OPTIMIZATION

We experimented with how precisely the band gap could be optimized. Here, we optimized the band gap to approximately 2.0, regardless of the crystal structure, and simultaneously minimized the formation energy. We conducted all methods in the same manner as mentioned in Section A.6. The results are shown in Table A.10.

Table A.10: Experiments with varying tolerance ranges for band gap optimization. The overall success rate is indicated by the probability of simultaneously satisfying three conditions: (A) the band gap is optimized within the target range, (B) the formation energy is below -0.5 eV, and (C) the crystal structure is valid. C is achieved when two criteria are met simultaneously: (a) all interatomic distances are greater than 0.5 Å, and (b) the structure is electrically neutral. S(Cry) and S(ALI) denote SMOACS utilizing the Crystalformer and ALIGNN models, respectively. We evaluate each of the proposed materials using all evaluation metrics, and the results are averaged over 512 samples.

| Target BG (eV) | method | success rate | (A)BG | (B)E$_f$ | (C)STR | (a) neut | (b) 0.5Å |
|---|---|---|---|---|---|---|---|
| 2.00 ±0.01 | S(Cry) | **0.234** | 0.355 | 0.594 | 0.781 | 0.984 | 0.781 |
| | S(ALI) | 0.010 | 0.012 | 0.811 | 0.830 | 0.910 | 0.830 |
| | TPE | 0.004 | 0.422 | 0.053 | 0.045 | 0.047 | 0.777 |
| | FTCP | 0.000 | 0.000 | 1.000 | 0.652 | 0.668 | 0.924 |
| 2.00 ±0.02 | S(Cry) | **0.230** | 0.340 | 0.543 | 0.742 | 0.977 | 0.742 |
| | S(ALI) | 0.037 | 0.043 | 0.795 | 0.803 | 0.893 | 0.803 |
| | TPE | 0.008 | 0.623 | 0.074 | 0.074 | 0.074 | 0.824 |
| | FTCP | 0.000 | 0.000 | 1.000 | 0.617 | 0.652 | 0.861 |
| 2.00 ±0.04 | S(Cry) | **0.277** | 0.418 | 0.566 | 0.738 | 0.984 | 0.738 |
| | S(ALI) | 0.082 | 0.092 | 0.820 | 0.838 | 0.914 | 0.838 |
| | TPE | 0.016 | 0.812 | 0.074 | 0.061 | 0.061 | 0.801 |
| | FTCP | 0.004 | 0.004 | 1.000 | 0.645 | 0.666 | 0.900 |
| 2.00 ±0.08 | S(Cry) | **0.238** | 0.410 | 0.562 | 0.730 | 0.980 | 0.730 |
| | S(ALI) | 0.111 | 0.119 | 0.816 | 0.803 | 0.912 | 0.803 |
| | TPE | 0.049 | 0.936 | 0.135 | 0.123 | 0.125 | 0.846 |
| | FTCP | 0.006 | 0.010 | 1.000 | 0.688 | 0.705 | 0.928 |
| 2.00 ±0.16 | S(Cry) | **0.316** | 0.516 | 0.570 | 0.750 | 0.988 | 0.750 |
| | S(ALI) | 0.193 | 0.225 | 0.832 | 0.820 | 0.910 | 0.820 |
| | TPE | 0.074 | 0.955 | 0.152 | 0.174 | 0.186 | 0.891 |
| | FTCP | 0.016 | 0.027 | 1.000 | 0.668 | 0.695 | 0.898 |

## A.11 GENERATION OF OXIDATION NUMBER PATTERNS

In SMOACS, realistic oxidation number patterns are generated based on the compositions of initial crystal structures. Here, we explain this using $RuN$ (mp-1009770). According to `icsd_oxidation_state` in PyMatGen, ruthenium (Ru) and nitrogen (N) can adopt oxidation numbers of $\{+2, +3, +4, +5, +6\}$ and $\{+1, +3, +5, -1, -2, -3\}$, respectively. Therefore, electrical neutrality in $RuN$ is achieved when the oxidation number combinations for Ru and N are $(+2, -2)$ or $(+3, -3)$. Consequently, when using $RuN$ (mp-1009770) as the initial structure, oxidation number combination patterns of $(+2, -2)$ and $(+3, -3)$ are obtained, and corresponding masks are generated for each.

To consider a broader range of oxidation number combinations, we utilized the intersection of oxidation numbers from "smact" and "icsd", as listed in Table A.7. It should be noted that even when generating oxidation number patterns from "smact" and "icsd", electrical neutrality is maintained by applying site-specific elemental constraints using the oxidation numbers in the "Ours" column of Table A.7.

## A.12 ADJUSTMENT OF PRIORITIES IN THE LOSS FUNCTION

For a fair comparison with TPE, which cannot prioritize each objective, we fixed the value of $\lambda$ to 1.0 in Equation 14. However, we considered it essential to investigate the effect of $\lambda$ in optimization and conducted experiments with various $\lambda$ values.

The results are shown in Table A.11. These experiments were conducted using SMOACS with Crystalformer, aiming for a band gap of 2.0 eV and optimizing crystal structures. We found that increasing $\lambda$, that is, placing greater emphasis on the formation energy during optimization, improved the success rate for formation energy. In addition, the probability that all interatomic distances are greater than 0.5 Å also increased. This is likely because emphasizing formation energy made it easier to avoid situations where atoms are too close together.

Table A.11: Experiments with varying $\lambda$ for a band gap target of 2.0 eV using SMOACS with Crystalformer. The overall success rate is indicated by the probability of simultaneously satisfying three conditions: (A) the band gap is optimized within the target range, (B) the formation energy is below -0.5 eV, and (C) the crystal structure is valid. Condition C is achieved when the following two criteria are met simultaneously: (a) all interatomic distances are greater than 0.5 Å, and (b) the structure is electrically neutral. We evaluate each of the proposed materials using all evaluation metrics, and the results are averaged over 128 samples.

| $\lambda$ | success rate | (A)BG | (B)$E_f$ | (C)STR | (a) neut | (b) 0.5Å |
|---|---|---|---|---|---|---|
| 0.040 | 0.086 | 0.492 | 0.164 | 0.617 | 0.945 | 0.617 |
| 0.200 | 0.188 | 0.477 | 0.328 | 0.656 | 0.984 | 0.656 |
| 1.000 | 0.336 | 0.453 | 0.727 | 0.852 | 1.000 | 0.852 |
| 5.000 | 0.266 | 0.320 | 0.836 | 0.875 | 0.992 | 0.875 |
| 25.000 | 0.086 | 0.102 | 0.859 | 0.875 | 1.000 | 0.875 |

## A.13 EXPERIMENTS WITH MODELS TRAINED ON OTHER DATASETS

We investigated whether optimization is feasible using models trained on datasets other than the MEGNet dataset. We used ALIGNN models on JARVIS dataset (Choudhary, 2021). The first experiment employed ALIGNN trained on the JARVIS DFT dataset. The second experiment utilized ALIGNN trained on the superconductivity dataset, SuperCon. In the first experiment using the JARVIS DFT data, we utilized models predicting the energy difference from the convex hull[4] and the bulk modulus[5]. The crystal structures were optimized to minimize the energy difference from the

---

[4] `https://figshare.com/articles/dataset/ALIGNN_models_on_JARVIS-DFT_dataset/17005681?file=31458658`

[5] `https://figshare.com/articles/dataset/ALIGNN_models_on_JARVIS-DFT_dataset/17005681?file=31458649`

convex hull less than $0.5$ eV while achieving the bulk modulus within the target range. The results are shown in Tables A.12 and A.13. Similar to the experiments conducted with models trained on the MEGNet dataset in the main text, optimizations using models trained on the JARVIS DFT data were successful. In the second experiment, we employed ALIGNN trained on the SuperCon dataset to predict the critical temperature (Tc)[6]. Since there is no formation energy prediction model trained on the superconductor dataset in JARVIS, we used the formation energy prediction model from the main text. Using these models, we optimized the crystal structures to maximize Tc above a certain temperature while minimizing the formation energy less than $-0.5$ eV. Specifically, the target temperatures for Tc were set to exceed the 0.901 and 0.963 quantiles of Tc within the dataset, corresponding to temperatures of 10 K and 15 K, respectively. The results are shown in Tables A.14. As with the experiments using models trained on the MEGNet dataset, optimization with models trained on the SuperCon data were successful.

Table A.12: Experiments for various bulk modulus using SMOACS with ALIGNN. The overall success rate is indicated by the probability of simultaneously satisfying three conditions: (A) the bulk modulus is optimized within the target range, (B) the energy difference from the convex hull less than $0.5$ eV, and (C) the crystal structure is valid. Condition C is achieved when the following two criteria are met simultaneously: (a) all interatomic distances are greater than $0.5$ Å, and (b) the structure is electrically neutral. We evaluate each of the proposed materials using all evaluation metrics, and the results are averaged over 512 samples.

| Target Bulk Modulus (GPa) | success rate | (A)BM | (B)$E_{hull}$ | (C)STR | (a) neut | (b) 0.5Å |
|---|---|---|---|---|---|---|
| $50.0 \pm 5.0$ | 0.020 | 0.090 | 0.123 | 0.512 | 0.578 | 0.512 |
| $75.0 \pm 5.0$ | 0.021 | 0.082 | 0.078 | 0.438 | 0.535 | 0.439 |
| $100.0 \pm 5.0$ | 0.016 | 0.102 | 0.066 | 0.395 | 0.510 | 0.395 |

Table A.13: Experiments for various bulk modulus while preserving perovskite structures using SMOACS with ALIGNN. The overall success rate is indicated by the probability of simultaneously satisfying three conditions: (A) the bulk modulus is optimized within the target range, (B) the energy difference from the convex hull less than $0.5$ eV, (C) the crystal structure is valid, and (D) approximating a valid perovskite structure. Criteria (A), (B), and (C) are consistent with those outlined in Table A.12. The Criteria (D) is consistent with that outlined in Table 3. We evaluate each of the proposed materials using all evaluation metrics, and the results are averaged over 512 samples.

| Target Bulk Modulus (GPa) | success rate | (A)BM | (B)$E_{hull}$ | (C)STR | (a) neut | (b) 0.5Å | (D)PS | (c) tole | (d) angles | (e) coord |
|---|---|---|---|---|---|---|---|---|---|---|
| $50.0 \pm 5.0$ | 0.072 | 0.309 | 0.326 | 0.939 | 1.000 | 0.939 | 0.525 | 0.525 | 1.000 | 1.000 |
| $75.0 \pm 5.0$ | 0.043 | 0.271 | 0.252 | 0.910 | 1.000 | 0.910 | 0.449 | 0.449 | 1.000 | 1.000 |
| $100.0 \pm 5.0$ | 0.020 | 0.178 | 0.176 | 0.928 | 1.000 | 0.928 | 0.416 | 0.416 | 1.000 | 1.000 |

Table A.14: Experiments for various Tc targets using SMOACS with ALIGNN. The overall success rate is indicated by the probability of simultaneously satisfying three conditions: (A) the Tc is optimized within the target range, (B) the formation energy is below -0.5 eV, and (C) the crystal structure is valid. Criteria C is consistent with that outlined in Table A.12. We evaluate each of the proposed materials using all evaluation metrics, and the results are averaged over 512 samples.

| Target Tc (K) | success rate | (A)Tc | (B)$E_f$ | (C)STR | (a) neut | (b) 0.5Å |
|---|---|---|---|---|---|---|
| $>10.0$ | 0.049 | 0.121 | 0.625 | 0.656 | 0.875 | 0.656 |
| $>15.0$ | 0.004 | 0.039 | 0.604 | 0.693 | 0.906 | 0.693 |

---

[6]https://figshare.com/articles/dataset/ALIGNN_models_on_JARVIS-DFT_dataset/17005681?file=38789199

## A.14 OPTIMIZATION WITH MARGINS BASED ON PREDICTOR ERROR

To ensure a fair comparison of optimization methods, we fixed the margin for all band gaps at 0.04 eV in the main text. However, in practical applications, it is conceivable to set the margin based on the predictor's prediction error. Here, we present the optimization results when the predictor's errors shown in Table 1 are used as margins. We performed optimizations regardless crystal structure and optimizations with preserving the perovskite structure, using the hyperparameters described in Sections A.5 and A.7, respectively. The results are shown in Tables A.15 and A.16. As shown in these Tables, FTCP has low expected values for satisfying the band gap value and maintaining the perovskite structure, and TPE has a low expected value for satisfying the formation energy; therefore, SMOACS has achieved the best results.

Table A.15: Experiments on optimizing for various targets of a band gap. We adopted the error values of each predictor in Table 1 as the margin for the band gap. The entries Cry, ALI, and F-Reg in the Predictor column correspond to the Crystalformer, ALIGNN, and regression branches of FTCP, respectively. Evaluation methods are based on those described in Table A.2.

| Target BG (eV) | method | Predictor | success rate | (A)BG | (B)$E_f$ | (C)STR | (a) neut | (b) 0.5Å |
|---|---|---|---|---|---|---|---|---|
| 0.50±0.20 | S(Cry) | Cry | 0.410 | 0.598 | 0.637 | 0.754 | 0.961 | 0.754 |
| 0.50±0.22 | S(ALI) | ALI | 0.336 | 0.387 | 0.855 | 0.867 | 0.941 | 0.867 |
| 0.50±0.20 | TPE | Cry | 0.109 | 1.000 | 0.289 | 0.305 | 0.336 | 0.945 |
| 0.50±0.44 | FTCP | F-Reg | 0.059 | 0.117 | 1.000 | 0.695 | 0.719 | 0.867 |
| 1.50±0.20 | S(Cry) | Cry | 0.504 | 0.645 | 0.727 | 0.828 | 0.984 | 0.828 |
| 1.50±0.22 | S(ALI) | ALI | 0.227 | 0.262 | 0.848 | 0.836 | 0.918 | 0.836 |
| 1.50±0.20 | TPE | Cry | 0.117 | 1.000 | 0.219 | 0.273 | 0.312 | 0.898 |
| 1.50±0.44 | FTCP | F-Reg | 0.039 | 0.039 | 1.000 | 0.695 | 0.727 | 0.875 |
| 2.50±0.20 | S(Cry) | Cry | 0.391 | 0.527 | 0.734 | 0.797 | 0.996 | 0.797 |
| 2.50±0.22 | S(ALI) | ALI | 0.238 | 0.270 | 0.844 | 0.781 | 0.914 | 0.781 |
| 2.50±0.20 | TPE | Cry | 0.062 | 0.961 | 0.148 | 0.172 | 0.188 | 0.891 |
| 2.50±0.44 | FTCP | F-Reg | 0.066 | 0.094 | 1.000 | 0.703 | 0.734 | 0.895 |

Table A.16: Experiments optimizing for various band gaps and while preserving a perovskite structure. We adopted the error values of each predictor in Table 1 as the margin for the band gap. The entries Cry, ALI, and F-Reg in the Predictor column correspond to the Crystalformer, ALIGNN, and regression branches of FTCP, respectively. Evaluation methods are based on those described in Table A.4.

| Target BG (eV) | method | Predictor | success rate | (A)BG | (B)$E_f$ | (C)STR | (a) neut | (b) 0.5Å | (D)PS | (c) tole | (d) angles | (e) coord |
|---|---|---|---|---|---|---|---|---|---|---|---|---|
| 0.50±0.20 | S(Cry) | Cry | 0.145 | 0.516 | 0.465 | 0.957 | 1.000 | 0.957 | 0.512 | 0.512 | 1.000 | 1.000 |
| 0.50±0.22 | S(ALI) | ALI | 0.191 | 0.516 | 0.473 | 1.000 | 1.000 | 1.000 | 0.449 | 0.449 | 1.000 | 1.000 |
| 0.50±0.20 | TPE | Cry | 0.039 | 1.000 | 0.492 | 0.914 | 0.914 | 1.000 | 0.258 | 0.258 | 1.000 | 1.000 |
| 0.50±0.44 | FTCP | F-Reg | 0.078 | 0.363 | 1.000 | 0.883 | 0.898 | 0.965 | 0.234 | 0.551 | 0.465 | 0.266 |
| 1.50±0.20 | S(Cry) | Cry | 0.227 | 0.566 | 0.527 | 0.980 | 1.000 | 0.980 | 0.594 | 0.594 | 1.000 | 1.000 |
| 1.50±0.22 | S(ALI) | ALI | 0.234 | 0.473 | 0.875 | 1.000 | 1.000 | 1.000 | 0.570 | 0.570 | 1.000 | 1.000 |
| 1.50±0.20 | TPE | Cry | 0.070 | 1.000 | 0.484 | 0.773 | 0.773 | 1.000 | 0.289 | 0.289 | 1.000 | 1.000 |
| 1.50±0.44 | FTCP | F-Reg | 0.039 | 0.129 | 1.000 | 0.867 | 0.883 | 0.961 | 0.258 | 0.531 | 0.465 | 0.281 |
| 2.50±0.20 | S(Cry) | Cry | 0.188 | 0.363 | 0.551 | 0.992 | 1.000 | 0.992 | 0.621 | 0.621 | 1.000 | 1.000 |
| 2.50±0.22 | S(ALI) | ALI | 0.238 | 0.379 | 0.961 | 1.000 | 1.000 | 1.000 | 0.641 | 0.641 | 1.000 | 1.000 |
| 2.50±0.20 | TPE | Cry | 0.062 | 1.000 | 0.508 | 0.656 | 0.656 | 1.000 | 0.320 | 0.320 | 1.000 | 1.000 |
| 2.50±0.44 | FTCP | F-Reg | 0.062 | 0.148 | 0.992 | 0.883 | 0.891 | 0.957 | 0.219 | 0.543 | 0.422 | 0.273 |

