# OpenReview forum: "Adaptive Constraint Integration for Simultaneously Optimizing Crystal Structures with Multiple Targeted Properties"
_ICLR.cc/2025/Conference — Submitted to ICLR 2025_

### Official Review · Reviewer_stg3 · 2024-10-31

**Soundness:** 2
**Presentation:** 3
**Contribution:** 2
**Rating:** 3
**Confidence:** 4

**Summary:**

This paper introduces SMOACS (Simultaneous Multi-property Optimization using Adaptive Crystal Synthesizer), a novel framework for optimizing crystal structures for multiple targeted properties while maintaining key constraints. Unlike traditional generative models, SMOACS utilizes state-of-the-art property prediction models and their gradients to optimize crystal structures via backpropagation directly. This approach allows adaptive constraint integration without model retraining, enabling SMOACS to propose materials optimized for properties like band gap and formation energy. SMOACS outperformed Bayesian optimization and generative models in experiments.

**Strengths:**

- Originality
  - This work proposes a novel problem of simultaneous multiple properties optimization.
- Quality
  - Extensive experiments are conducted and the DFT is performed to evaluate the generation.
- Clarity
  - The paper is well presented with a clear explanation of the methods.
- Significance
  - The problem is significant for new material discovery.

**Weaknesses:**

- The novelty of the method
   - To optimize the property of materials with the gradient of property prediction methods is not new. It was proposed in section 5.3 of [1]
- The evaluation
  - only two properties are optimized, how about more properties like the mechanic properties? How to handle the conflict between different property prediction models.
  - the success rate is poor.
- The significance of the parameter $\lambda$ in Eq.14
  - How is the strength parameter chosen? It seems to be important to balance the optimization between target property and formation energy. Besides, for more properties, should we use different parameters? This is important since it is not only involved in the training but also the optimization of crystal structures.


[1] Tian Xie, ICLR2022, CRYSTAL DIFFUSION VARIATIONAL AUTOENCODER FOR PERIODIC MATERIAL GENERATION

**Questions:**

- parameter significance.
  - see weakness
- the difference between this method and the method used in CDVAE [1].

[1] Tian Xie, ICLR2022, CRYSTAL DIFFUSION VARIATIONAL AUTOENCODER FOR PERIODIC MATERIAL GENERATION

---

> ### Author Response · Authors · 2024-11-20
> **Response to your concerns and questions (1/3)**
>
> We sincerely appreciate your valuable feedback and insightful questions regarding our manuscript. Your comments have provided significant guidance to enhance our research, and we are grateful for your thoughtful review.
>
> The latexdiff with the updated paper is available in the Supplementary Material at the following URL: https://openreview.net/attachment?id=NVKwjCIAAX&name=supplementary_material
>
> Below, we address the weaknesses and questions you have raised.
>
>
>
> —---------
> > Weakness1 :The novelty of the method To optimize the property of materials with the gradient of property prediction methods is not new. It was proposed in section 5.3 of [1]
>
> Thank you for your comment regarding the novelty of our method. While it is true that both our approach and the model proposed in Section 5.3 of the CDVAE paper (referred to as CDVAE-5.3) utilize gradients for optimization, there are three significant advantages.
>
> Firstly, CDVAE-5.3, may experience a decrease in predictive accuracy due to its inability to utilize the most accurate models. It is important to note that CDVAE-5.3 uses a latent space for exploring new materials, a technique very similar to FTCP, which we used as a benchmark in SMOACS. A crucial difference lies in the use of latent variables for property prediction. As discussed in Section 4.1, CDVAE-5.3, similar to FTCP, may experience a decrease in predictive accuracy due to its inability to utilize the most accurate models. SMOACS circumvents this limitation, as demonstrated in Table 1.
>
> Secondly, the entanglement of different crystal components within the latent space can complicate the preservation of specific crystal structures. FTCP struggles with this issue, as shown in Table 3. CDVAE-5.3 may also struggle with this issue. In contrast, SMOACS directly handles crystal structures, allowing for the optimization while maintaining their specific configurations, as outlined in Section 3.3.
>
> Thirdly, CDVAE-5.3 is unable to ensure electrical neutrality and preserve specific crystal structures because generative models are probabilistic models. While FTCP also fails (and CDVAE-5.3 may fail) to maintain these aspects, as shown in Table 3, SMOACS successfully performs conditioning-based optimization, as demonstrated in Sections 4.3 and 4.4. This capability is crucial in materials design outlined in our Introduction.
>
>
> We appreciate your feedback and have realized the necessity of clarifying these distinctions. We have now included a new paragraph to show the novelty in Introduction as follows:
> "SMOACS is the first method that directly optimizes the space of crystal structures using a gradient-based approach. We achieve this by making the entire crystal structure differentiable, which involves decomposing it into various components and representing atomic species as atomic distributions. Unlike traditional methods that convert crystal structures into latent variables—thereby entangling their elements—our approach maintains the independence of each component. This independence facilitates the preservation of crystal structures and ensures electrical neutrality by precisely specifying the atoms at each site. Furthermore, unlike generative models that probabilistically generate materials satisfying certain conditions, our method can inherently guarantee electrical neutrality and the preservation of crystal structures. Moreover we can add additional constraints as long as they are differentiable."

---

> > ### Author Response · Authors · 2024-11-20
> > **Response to your concerns and questions (2/3)**
> >
> > > Weakness2 :The evaluation. only two properties are optimized, how about more properties like the mechanic properties? How to handle the conflict between different property prediction models. the success rate is poor.
> >
> > Regarding your point about optimizing only two properties, we would like to clarify that in Section 4.2, we simultaneously optimize both the band gap and the formation energy. Furthermore, in Sections 4.3 and 4.4, following Equation 17, we optimize three properties simultaneously by including the tolerance factor in addition to the band gap and formation energy.
> > In comparison, FTCP[2] and CDVAE-5.3[1] optimize only a single property. In Figure 5 of the MatterGen[3] paper, they explore stability (and novelty, S.U.N.) alongside either only the band gap or only the bulk modulus; thus, the number of properties being optimized simultaneously is at most two. Considering this, the simultaneous optimization of three properties (two properties plus the tolerance factor) in our method represents a significant advancement.
> > The number of factors being explored, our research is on par with other studies. MatterGen explores five factors: stability (S.U.N.), magnetism, limitation of atomic species (composition formula and supply chain risks), and 14 types of crystal structures (space symmetry). In our paper, we also explore five factors: in addition to stability, we consider eight types of band gaps (Table A.3, A.4, and A.5), crystal structures, tolerance factor, and electrical neutrality.
> >
> > On the other hand, we understand your concerns.  Regarding mechanical properties such as Young's modulus, We are currently investigating whether we can conduct experiments using models trained on other datasets.
> >
> > When addressing conflicts among different property prediction models, the optimization outcomes of SMOACS are dependent on the initial values. Poor initial values may lead to conflicts due to being trapped in local minima.  Since overcoming local minima is challenging, we can adopt a strategy of simultaneously optimizing a large number of samples (e.g. 2048) to find better initial values.
> > Concerning the success rate being poor, we believe that the low success rate is not a significant issue in practice. Our method allows us to perform simultaneous optimization of 2,048 samples in a few minutes using a single A100 GPU. Therefore, even with a success rate around 10%, we can efficiently collect a substantial number of successful samples.  Your comments have underscored the importance of incorporating this information into the main body of our text. Consequently, we have added the following sentences:
> > “SMAOCS can easily scale this computation and can optimize 2,048 samples simultaneously in just a few minutes using a single A100 GPU. This allows us to repeat the optimization process multiple times, enabling us to obtain a large number of successful optimization samples. “ in Section 4.1
> >
> > —---------
> > > Weakness 3 :The significance of the parameter λ in Eq.14. How is the strength parameter chosen? .... Besides, for more properties, should we use different parameters? This is important since it is not only involved in the training but also the optimization of crystal structures.
> >
> > Thank you for highlighting the significance of the parameter λ in Equation 14. In our study, we set λ = 1 to align with the TPE, which cannot prioritize the weighting of material properties. This choice facilitates fair comparisons between our method and TPE under the same conditions.
> > One of the strengths of our approach is the ability to adjust priorities using λ. The selection of λ is indeed crucial and should be tailored based on the specific problem setting and which material properties are being prioritized. By adjusting the value of λ, we can balance the optimization between the target property and the formation energy to suit different objectives.
> > Recognizing the importance of this parameter, we have conducted an investigation into the effects of varying λ. We found that increasing λ, that is, placing greater emphasis on the formation energy during optimization, improved the success rate for formation energy. In addition, the probability that all interatomic distances are greater than 0.5 Å also increased. This is likely because emphasizing formation energy made it easier to avoid situations where atoms are too close together. The results of this analysis have been added to Appendix A.12 in the revised manuscript.

---

> > > ### Author Response · Authors · 2024-11-20
> > > **Response to your concerns and questions (3/3)**
> > >
> > > > Questions: parameter significance. see weakness. the difference between this method and the method used in CDVAE [1].
> > >
> > > For the first question, I have addressed it in Weakness 1, and for the next question, I have responded in Weakness 4.
> > >
> > > —---------
> > >
> > > I hope that all of your concerns are resolved. Thank you very much for your attention!
> > >
> > > [2] Ren, Zekun, et al. "An invertible crystallographic representation for general inverse design of inorganic crystals with targeted properties." Matter 5.1 (2022): 314-335.
> > >
> > > [3] Zeni, Claudio, et al. "Mattergen: a generative model for inorganic materials design." arXiv preprint arXiv:2312.03687 (2023).

---

> ### Author Response · Authors · 2024-11-22
> **Additional experiment regarding to Weakness2**
>
> > Weakness2: The evaluation. only two properties are optimized, how about more properties like the mechanic properties? How to handle the conflict between different property prediction models.
>
> Thank you for your valuable suggestion regarding the evaluation of additional properties. In response to your advice, we have expanded our experiments to include new properties, as summarized in Section A.13. To evaluate properties distinct from the electronic property (band gap), we conducted experiments using models trained on two datasets, which facilitate the optimization of mechanical properties (bulk modulus) and superconducting transition temperatures.  For each of these properties, we conducted experiments using ALIGNN models trained accordingly. Although the success rate appears lower, we do not regard this as a significant concern. SMOACS demonstrates high efficiency, capable of optimizing thousands of samples within a few ~ 30 minutes using a single A100 GPU, as detailed in Section 4.2.
>
> Our study now optimizes six properties in total: bulk modulus, the energy difference from the convex hull, critical temperature of superconductors, band gap, formation energy, and tolerance factor. This expansion allows us to address a broader range of properties, including mechanical properties, thereby providing a more comprehensive evaluation of our approach. Additionally, we add the following sentence to the first paragraph of Section 4 to ensure readers are aware:
>
>  “For experiments utilizing models trained on datasets other than MEGNet, please refer to Section A.13.”

---

> > ### Comment · Reviewer_stg3 · 2024-11-27
> >
> > Thank you for your response, I have read the rebuttal carefully and the questions are as follows:
> > - Why CDVAE and FTCP can not utilize the most accurate models? The key idea of all these models including your SMAOCS is utilizing the gradients.
> > - I still disagree with the claim that "multiple targeted properties can be optimized with SMAOCS" for two reasons.
> >   - the experiments only show the optimization of the tolerance factor, band gap, and formation energy. However, the success rate is quite poor.
> >   - Randomly initializing the crystal structure is not a good way to handle the conflict between different property prediction models.
> >
> > Thank you again for your clarification. However, I still take this idea as incremental work. I encourage the author to explore a better way to handle the conflict of different property prediction models and improve the success rate.

---

> ### Author Response · Authors · 2024-12-03
> **Regarding the success rate and random initialization**
>
> Thank you very much for your insightful comments. Regarding the success rate, we would greatly appreciate it if you could specify the target success rate you consider appropriate for this problem, along with your reasoning. Additionally, if you are aware of any better methods than our initialization approach (please note that our method is not completely random but is based on physical principles), we would greatly appreciate it if you could share the details, as it would be valuable for our future research. We will address your other concerns in a subsequent response.

---

> ### Author Response · Authors · 2024-12-04
> **Addressing your concerns (1/3)**
>
> We appreciate your important comments.
>
> > 1. Why CDVAE and FTCP can not utilize the most accurate models? The key idea of all these models including your SMAOCS is utilizing the gradients.
>
> We would like to first clarify that FTCP does not use gradients. It employs the sampling mechanism of the VAE to sample around data close to the target properties, as described in Section A.6. As far as we know, there is no method applying gradient-based optimization to FTCP.
>
> [__Why cannot use most accurate models__]: Firstly, authors of CDVAE and FTCP propose to predict physical properties from latent variables; On the other hand, the most accurate models such as ALIGNN and Crystalformer require crystal structures as inputs, which is distinctly different from latent variables. Therefore, CDVAE and FTCP cannot use such models.
>
> [__On the use of gradients__]:Secondly, while both our approach and CDVAE use gradients, our approach does not require the training of a generative model such as CDVAE and therefore goes far beyond the contributions of CDVAE. We acknowledge that, in principle, it might be possible to connect models such as Crystalformer following the decoder of CDVAE (it was not originally proposed in CDVAE paper[1]). However, our method only requires an off-the-shelf, web-available, pre-trained ALIGNN, which differs from CDVAE that requires the training of a generative model. CDVAE is trained on the perovskite structure dataset, Peov-5, for generating perovskite structures, and this is necessary regardless of whether ALIGNN is used or not. In contrast, SMOACS does not require such retraining and can generate perovskite structures even using only the web-available ALIGNN trained on various crystal structures.
>
> [__Difference between FTCP and SMOCAS__]: Thirdly, as detailed in “Response to your concerns and questions (1/3) Weakness 1,” the entanglement of different crystal components within the latent space complicates the preservation of specific crystal structures. FTCP (and CDVAE-5.3 may also) struggles to maintain electrical neutrality and specific configurations, as evidenced in Table 3. In contrast, SMOACS directly manages crystal structures rather than latent space, enabling optimization while preserving specific crystal structures, as outlined in Section 3.3, and successfully performs conditioning-based optimization. This capability is crucial for materials design, as discussed in our Introduction.
>
> In conclusion, FTCP and CDVAE utilize latent variables for prediction, thereby they can not  adopt  the most accurate models such as ALIGNN as a predictive model. A significant contribution of SMOACS lies in its ability to incorporate off-the-shelf models without the necessity for additional training. By directly managing crystal structures, enabling optimization while preserving specific crystal structures and electrical neutrality. Thus, although both SMOACS and CDVAE employ gradient-based optimization, there are critical differences.

---

> ### Author Response · Authors · 2024-12-04
> **Addressing your concerns (2/3)**
>
> > 2. I still disagree with the claim that “multiple targeted properties can be optimized with SMAOCS” for two reasons.
> The experiments only show the optimization of the tolerance factor, band gap, and formation energy. However, the success rate is quite poor.
>
> Thank you very much for your comment.
>
> [__Difficulty in new benchmarking__] Firstly, the problem we are tackling involves exploring unknown territories and may not have solutions, unlike machine learning benchmark tasks with known correct answers; therefore, a low success rate is inevitable. For example, developing perovskite solar cells (Sections 4.3 and 4.4) present challenges, including the incorporation of lead and relatively low power conversion efficiencies (strongly related to the bandgap optimization we are pursuing), and may not have ideal materials. Consequently, their widespread practical implementation has remained limited.
>
> [__Comparison with other methods__]  Secondly, although we are tackling more challenging problems than other studies, our success rate is not significantly different. MatterGen[3], in Figure 4, performs energy minimization searches under the condition of a given crystal structure (i.e., solving two conditional problems), addressing a total of 14 conditional problems related to crystal structures. However, out of these 14 cases, seven have success rates below 20%, with some even falling below 10%. In our experiments involving crystals having a similar number of atoms (Section 4.3), we achieved a success rate of approximately 10 ~ 20% under five conditions: three optimization targets, electrical neutrality, and preservation of the crystal structure. In Section 5.3 of CDVAE, they calculate scores after selecting the best out of 10 samples; thus, when aligning their evaluation method with ours, their success rate would be roughly 1/10, falling below 10%. Therefore, compared to other studies, our success rate is by no means low.
>
> [__Practical point of view__]  Thirdly, we believe practical usefulness in materials design is more important than the absolute success rate. The ultimate goal is to synthesize promising materials, and if machine learning can propose promising candidates faster than synthesis, the method is valuable. As we showed in the “Additional experiment regarding Weakness 2,” we are able to obtain approximately 500 successful candidates per hour using a single GPU. Thus, the success rate of SMAOCS is sufficient for practical purposes.
>
> [__Ensuring electrical neutrality and preserving specific crystal structures__] Fourthly, we have solved problems where existing methods such as FTCP and TPE failed, achieving accuracy sufficient for practical materials design. This demonstrates that we have, for the first time, solved a problem that was previously unsolved, which holds great significance. Particularly, the difficulty of calculating electrical neutrality discussed in Section 4.4 is noteworthy. This indicates that while materials proposed by methods other than SMOACS may achieve the desired properties, they cannot verify the electrical neutrality essential for realistic materials. In contrast, SMOACS exhibits a significant advantage by being able to bypass this limitation.
>
> As discussed above, although the success rate may appear low compared to benchmarks such as ImageNet, we conclude that SMAOCS is sufficiently effective.

---

> ### Author Response · Authors · 2024-12-04
> **Addressing your concerns (3/3)**
>
> > 3. Randomly initializing the crystal structure is not a good way to handle the conflict between different property prediction models.
>
> We apologize for the insufficient description given in section 3.2. We would like to emphasize that while our initialization is not done in a completely random way.
>
> [__physics based initialization__] Firstly, our initialization method is not entirely random; it is based on physics domain knowledge. In Section 4.2, we select initial structures from the MEGNet dataset, mimicking common techniques such as element substitution, where new materials are derived from existing crystal structures. In Section 4.3, we use randomly distorted perovskite structures with specified atomic coordinates and charges to ensure they are likely to exist. This approach effectively leverages physics knowledge and is reasonable.
>
> [__local minima are natural from the physical point of view__] Secondly, from a condensed matter physics perspective, conflicts between different property predictions are natural and inherently difficult to overcome. Although the crystal structure search space appears vast, practical materials occupy limited regions. Only a limited number of crystal structure types exist, and achieving specific properties can be challenging depending on the structure (e.g., the face-centered cubic structure family is typically metallic with a bandgap of 0 eV). Consequently, the search space is sparse with many regions where property gradients are zero (e.g., in the region of face-centered cubic structure family, gradient toward >0 eV might not exist), leading to conflicts and local minima. Overcoming these local minima is difficult and may result in unstable structures outside the data distribution. We believe this characteristic contributes to the low success rates observed in FTCP, TPE and SMAOCS.
>
> [__physically appropriate strategy__] Thirdly, given the sparsity and gradient issues, starting from known crystal structures and optimizing locally is the most efficient strategy to overcome these conflicts. Our approach aligns with this by initiating optimization based on real crystal structures.
>
> As discussed, our initialization method is efficient and utilizes physics knowledge. Your feedback has highlighted the need to improve our writing style. We have revised the opening sentence of Section 3.2 and included references to the Appendix for further details:
> “*In SMOACS, the initial structures for the optimization process are obtained from two sources: an  existing dataset and self-generated structures based on typical perovskites with modified lattice constants (for details, see Section A.6 and A.7).*”
>
> > 4. However, I still take this idea as incremental work. I encourage the author to explore a better way to handle the conflict of different property prediction models and improve the success rate.
>
> As discussed above, our methodology efficiently addresses the conflicts using a physics-based perspective approach. Furthermore, we tackle challenging problems for which solutions may not exist, making a lower success rate inevitable; indeed, other approaches often fail under these conditions. The success rate of our method is sufficient for materials design and remains adequately practical in its current state.
>
> Since overcoming local minima is challenging from the physical point of view, improving the success rate can be achieved by further exploring the vicinity of promising initial values rather than adopting better optimization techniques. For example, by using the results from the first SMOACS run to perturb promising initial values, we can further explore the surrounding area of promising solutions[4].
>
> I hope that all of your concerns are resolved. Thank you very much for your time and consideration!
>
> [4] Fujii, Akihiro, et al. “Enhancing Inverse Problem Solutions with Accurate Surrogate Simulators and Promising Candidates.” arXiv preprint arXiv:2304.13860 (2023).

---

### Official Review · Reviewer_p9py · 2024-11-02

**Soundness:** 3
**Presentation:** 2
**Contribution:** 2
**Rating:** 5
**Confidence:** 3

**Summary:**

The paper proposes an optimization-based method (SMOACS) for materials discovery which aims to optimize (possibly multiple) target properties that are predicted by off-the-shelf predictive ML models. The optimization variables are the axes of a unit cell, the 3D position of N atoms in the cell, probability vectors over K possible elements for each atom, and a D-dimensional probability vector which determines the “oxidation pattern” by essentially selecting a pattern from D templates. A key feature of the method is that, for any given oxidation pattern, the atoms populating the N sites are restricted to adhere to that particular pattern in order to ensure electrical neutrality. With this constraint in place, the method is essentially a first-order gradient-based optimization algorithm with gradients obtained by auto-differentiating the predictive ML models w.r.t. their inputs.

**Strengths:**

The proposed approach to directly optimize the representation of the material to maximize certain (ML-predicted) properties is interesting. It makes it straightforward to use any available predictive model, as long as it is differentiable w.r.t. its input. It also makes it easy to constrain the optimization algorithm to respect e.g. structure constraints by simply turning off the optimization of some variables (illustrated in the paper) or by adding explicit constraints to the optimizer (not illustrated in the paper).

**Weaknesses:**

* In the numerical evaluation the authors show a high “success rate” for SMOACS, which is a metric tailored to the specific properties that are optimized and/or hard-coded into the proposed methods. However, it was not clear to me from the description to which extent the alternative methods (optimization-based TPE and generative FTCP) explicitly target the same properties. It’s mentioned in section 4.4 that TPE includes a term in the objective function for optimizing electric neutrality (except for the results in sec 4.4) but it’s not clear how this is implemented. Nor is it clear how the different terms in the objective function for the different methods are weighed against each other. Since the alternative methods seem to outperform SMOACS on some metrics (e.g. TPE shows good results on BG and FTCP on E_f in Table 2), I wonder how much a better tuning of the these methods could improve the “success rate”. All in all, the numerical evaluation leaves me wondering if this is really a fair comparison.

* Compared with the generative approach (FTCP), SMOACS seems to struggle to find (meta-)stable, and thus synthesizeable, materials as measured by formation energy and inter-atomic distances. Furthermore, stability of a material is a relative property not directly computable from formation energy—for a material to be stable it needs to be in a more favorable energy state than competing phases---making it even harder to say that the materials found by SMOACS are potentially synthesizable. I would say that this is a limitation of the purely optimization-based approach (which only cares about optimizing the target properties of interest) compared with the generative approach (which also tries to generate structures that are “in distribution”, i.e. similar in some sense to the training data samples). I would have liked to see a more in-depth discussion about this limitation (??) of the proposed method.

* The authors deliberately use worse property prediction models for the generative approach (FTCP). Although I appreciate the point that the authors want to make, that SMOACS easily can make use of any (differentiable!) SOTA predictive model which is not always the case for alternative approaches, I would have liked to see comparison with a generative model based on SOTA prediction networks. In particular considering that there are many training-free methods available for conditional generation (see e.g. https://arxiv.org/abs/2306.17775 and the references therein). I am not aware of any such methods specifically for crystal generation (although I would be surprised if no such methods exists), this line of research should at least be acknowledged.

* SMOACS is based on differentiating through the property prediction model w.r.t. its input. It is not clear how this handled non-differentiability of the used model. For instance, GNN-based methods typically construct a graph based on atomic distances, which means that the graph itself might change during the optimization phase of SMOACS (since you “move the atoms around”).

* SMOACS ensures electric neutrality by restricting the search to structures that have the same oxidation pattern as one of a given number of D templates. This seems to restrict the optimization quite a bit, and the choice of D is not discussed in the paper.

* In general I found the paper quite hard to read. Specifically, it lacks a clear and concise formulation of the problem and description of the proposed method early in the paper.

**Questions:**

Why is it computationally infeasible to calculate electrical neutrality? Enumerating all possible combinations of oxidation numbers will of course be combinatorial, but simply computing the electric neutrality for a given structure should be straightforward, no?

---

> ### Author Response · Authors · 2024-11-20
> **Response to your concerns and questions (1/4)**
>
> We sincerely appreciate your valuable feedback and insightful questions regarding our manuscript. Your comments have provided significant guidance to enhance our research, and we are grateful for your thoughtful review.
>
> The latexdiff with the updated paper is available in the Supplementary Material at the following URL: https://openreview.net/attachment?id=NVKwjCIAAX&name=supplementary_material
>
> Below, we address the comments, weaknesses and questions you have raised.
>
>
> —---------
> > Strengths : The proposed approach to directly optimize the representation of the material to maximize certain (ML-predicted) properties is interesting. … by adding explicit constraints to the optimizer (not illustrated in the paper)
>
> Thank you for your valuable comments. Upon reading the final comment in the Strengths section, which states "by adding explicit constraints to the optimizer (not illustrated in the paper)," we realized that this part may be unclear. We have included the following paragraph in the Introduction to describe strengths beyond those you have pointed out.
>
>
> “SMOACS is the first method that directly optimizes the space of crystal structures using a gradient-based approach. We achieve this by making the entire crystal structure differentiable, which involves decomposing it into various components and representing atomic species as atomic distributions. Unlike traditional methods that convert crystal structures into latent variables~\citep{REN2022314}—thereby entangling their elements—our approach maintains the independence of each component. This independence facilitates the preservation of crystal structures and ensures electrical neutrality by precisely specifying the atoms at each site. Furthermore, unlike generative models that probabilistically generate materials satisfying certain conditions, our method can fundamentally guarantee electrical neutrality and the preservation of crystal structures. Moreover we can add additional constraints as long as they are differentiable.”
>
>
> —---------
> > Weakness1 : In the numerical evaluation the authors show a high “success rate” for SMOACS, …..the numerical evaluation leaves me wondering if this is really a fair comparison.
>
> Thank you for your comments and insights regarding the numerical evaluation and comparison of the methods presented in our manuscript. We value your attention to the detailed implementation and the fairness of our comparisons.
> In response to your concerns, we'd like to first clarify that the implementation details for both the FTCP and TPE methods are fully documented in Sections A.6 and A.7 of the supplementary material. Concerning the fairness of the comparison, we have made deliberate efforts to ensure that both SMOACS and the alternative methods are evaluated under comparable conditions. As indicated in the last paragraph of Section A.4, various parameters were adjusted for the FTCP. With respect to TPE, we designed an objective function analogous to that used in SMOACS and optimize all objectives equally. To ensure consistency and fairness in comparison, we set  𝜆 = 1.0 in Equation 14 for SMOACS, preventing any advantage that could arise from adjusting weights of the objective functions.
> Additionally, the bandgap margin of 0.04 eV mentioned is not arbitrarily designed to favor SMOACS but is rather based on the typical discrepancies between DFT calculations and experimental values. Note that the superiority of SMOACS holds across various margin values as shown in Table A.10.

---

> ### Author Response · Authors · 2024-11-20
> **Response to your concerns and questions (2/4)**
>
> —---------
> > Weakness 2: Compared with the generative approach (FTCP), SMOACS seems to struggle to find (meta-)stable, and thus synthesizeable, materials as measured by formation energy and inter-atomic distances. Furthermore, stability of a material is a relative property not directly computable from formation energy—for a material to be stable it needs to be in a more favorable energy state than competing phases---making it even harder to say that the materials found by SMOACS are potentially synthesizable. I would say that this is a limitation of the purely optimization-based approach (which only cares about optimizing the target properties of interest) compared with the generative approach (which also tries to generate structures that are “in distribution”, i.e. similar in some sense to the training data samples). I would have liked to see a more in-depth discussion about this limitation (??) of the proposed method.
>
>
> Thank you for your comments and insights regarding the stability of materials. We acknowledge that SMOACS appears to struggle more than generative models in finding (meta-)stable materials. However, our primary objective is not solely focused on stability. For instance, FTCP often fails to satisfy the specified structure or bandgap requirements.
> We also agree that considering the possibility of other phases with different composition ratios is crucial in examining practical stability of materials. However, calculating all possible competing phases and determining which ones are the most stable is a very challenging task for both SMOACS and generative models. In fact, Fig. 2(b) of MatterGen's paper[1] reveals that a relatively small proportion of those that are generated by the generative model was actually the most stable.
> From this perspective, taking account of the practical stability of materials remains as a future challenge for both SMOACS and generative models.
>
> We appreciate your feedback and will include a following discussion about the stability in the Conclusion section in the revised manuscript.
> “However, simply minimizing the formation energy does not guarantee that the material is synthesizable; it is necessary to carefully investigate the stability of competing phases.”
>
>
> —---------
> > Weakness 3: The authors deliberately use worse property prediction models for the generative approach (FTCP). … I would have liked to see comparison with a generative model based on SOTA prediction networks.
>
> As you have pointed out, in our optimization experiments using FTCP (e.g., Table 2), we employed a property prediction model with lower accuracy. However, the accuracy of the property prediction model does not influence the optimization results presented in Tables 2, 3, and 4, ensuring that our experiments remain fair. In these experiments, we optimized the crystals so that the predicted property values from the property prediction model fall within specified ranges (e.g., 2.00 ± 0.04 eV), rather than relying on the ground truth property values. In essence, we treated the predicted values from the property prediction model as virtual true values for the purpose of optimization. This facilitates a comparison between the optimization algorithms of generative models and SMOACS, under the assumption that each employs an ideal prediction model.
> Therefore, whether we use a state-of-the-art (SOTA) model or a less accurate one, the prediction accuracy does not impact the experimental outcomes. This approach ensures fairness irrespective of the prediction model employed.
> Nonetheless, we understand that this aspect might not have been clear in our paper. To address this, we have revised sentence in the introductory paragraphs of Sections 4.2 and add sentence on
> second to last paragraph Section 4.3, such as "We evaluated the structures based on the band gap and formation energy values predicted by their respective predictors."

---

> > ### Author Response · Authors · 2024-11-20
> > **Response to your concerns and questions (3/4)**
> >
> > —---------
> > > Weakness 3.5: ... In particular considering that there are many training-free methods available for conditional generation (see e.g. https://arxiv.org/abs/2306.17775 and the references therein). I am not aware of any such methods specifically for crystal generation (although I would be surprised if no such methods exists), this line of research should at least be acknowledged.
> >
> > Thank you for your valuable comment and for bringing a highly relevant paper to our attention. We have appropriately acknowledged this line of research by adding a discussion of this study to the Related Work section.
> > To the best of our knowledge, there are no existing training-free methods specifically for conditional crystal generation. This absence highlights one of the key contributions of our work. As indicated in the MatterGen paper[1] and our Related works section, the community has primarily focused on generating stable materials, with little attention to property-conditioned generation. Furthermore, in materials science, conditional generation typically requires integration with a property prediction model, which may be one of the reasons such methods have not been developed for crystals.
> > Even if a technique such as TDS[2] combined with a state-of-the-art prediction model and MatterGen were used to construct a conditional generation model without retraining for crystals—which would itself be a novel and significant contribution—it would still be challenging to ensure electrical neutrality and specific crystal structures.
> > Our method inherently satisfies these conditions, providing a distinct advantage. As pointed out in Section 4.5, verifying electrical neutrality in large systems is difficult, making it challenging to calculate and enforce electrical neutrality for each sampling in methods like TDS. We believe that our approach offers significant benefits in this regard.
> >
> >
> > —---------
> > > Weakness 4: SMOACS is based on differentiating through the property prediction model w.r.t. its input. It is not clear how this handled non-differentiability of the used model. For instance, GNN-based methods typically construct a graph based on atomic distances, which means that the graph itself might change during the optimization phase of SMOACS (since you “move the atoms around”).
> >
> > Thank you for your insightful comment. As you have correctly noted, the optimization process in SMOACS can lead to changes in the nearest neighbor atoms, necessitating updates to the graph during optimization. In optimization using ALIGNN, we reconstruct the graph multiple times based on the evolving structures during the optimization; this process is detailed in Section A.6 of our manuscript. In light of your comment, we recognized the importance of including this explanation in the main text as well. Accordingly, we have added this information to the final paragraph of Section 3.2.
> >
> >
> >
> > —---------
> > > Weakness 5: SMOACS ensures electric neutrality by restricting the search to structures that have the same oxidation pattern as one of a given number of D templates. This seems to restrict the optimization quite a bit, and the choice of D is not discussed in the paper.
> >
> > For selecting D, our method proceeds as follows: we randomly select crystal structures from the dataset and directly use the oxidation state combinations of these structures as D templates. An explanation of this procedure is provided at the beginning of Section 3.2, with further detailed descriptions in Section A.11. In response to your comment, we will revise the sentence in Section 3.2 to be readily understandable as follows: "These crystal structures must satisfy electrical neutrality and generate $D$ oxidation number patterns based on the compositions of initial crystal structures."
> >
> >
> > —---------
> > > Weakness 6: In general I found the paper quite hard to read. Specifically, it lacks a clear and concise formulation of the problem and description of the proposed method early in the paper.
> >
> > Thank you for your very helpful comment. We appreciate your feedback regarding the clarity of our paper. In response, we have added a new first paragraph that describes the formulation of the problem and our method as follows:
> > “We address the challenge of simultaneously optimizing multiple material properties while preserving specific crystal structures and ensuring electrical neutrality. To achieve this, we have developed a methodology that leverages property prediction models and their gradients to facilitate the discovery of materials with multiple desired properties. This approach allows for the adaptive application of constraints, such as electrical neutrality and specific crystal structures, without necessitating retraining. As a result, our method enables the optimization of large atomic configurations to obtain specific properties while ensuring electrical neutrality and preserving specific crystal structures.”

---

> > > ### Author Response · Authors · 2024-11-20
> > > **Response to your concerns and questions (4/4)**
> > >
> > > > Question : Why is it computationally infeasible to calculate electrical neuatrality? Enumerating all possible combinations of oxidation numbers will of course be combinatorial, but simply computing the electric neutrality for a given structure should be straightforward, no?
> > >
> > > Thank you for your question. Indeed, calculating the electrical neutrality for a given combination of oxidation numbers is straightforward, as it simply involves summing the oxidation states over all atomic sites. In many cases, this total sum does not equal zero. In such cases, there may exist other combinations of oxidation numbers that result in a net zero charge. To ensure electrical neutrality in a crystal, it is necessary to consider all possible combinations of oxidation numbers and identify those where the total sum equals zero. One of the advantages of our method is that it can skip this heavy calculation during optimization.
> > >
> > > —----------
> > >
> > > I hope that all of your concerns are resolved. Thank you very much for your attention!
> > >
> > >
> > > [1] Zeni, Claudio, et al. "Mattergen: a generative model for inorganic materials design." arXiv preprint arXiv:2312.03687 (2023).
> > >
> > > [2] Wu, Luhuan, et al. "Practical and asymptotically exact conditional sampling in diffusion models." Advances in Neural Information Processing Systems 36 (2024).

---

> > ### Comment · Reviewer_p9py · 2024-11-21
> >
> > Regarding Weakness 3: Ok, but to me this still seems like a strange way of evaluating the performance of the models. Since the "success rate" is based on absolute numbers (e.g. band gap within +-0.04 eV), if you use a predictive model with a very high variance in your evaluation, wouldn't this result in a worse success rate simply due to mismatch in the range of predicted values and the target range? If you want to use the predictive model to evaluate the success, wouldn't it make more sense to have a relative target range based on the variance of the predictor?

---

> > > ### Author Response · Authors · 2024-11-23
> > > **Add experiments using the prediction errors of the predictors as margins for band gap target**
> > >
> > > Thank you for your insightful comment. Although we still think the setting of our experiments is reasonable in the light of fair comparison, we understand that, in practical applications, it is necessary to consider the variance of the predictor, such as adjusting the target ranges to ±0.20 eV and ±0.44 eV for Crystalformer and FTCP respectively due to their band gap prediction errors of 0.20 eV and 0.44 eV. Therefore, we have added experiments in Appendix A.14 using the prediction errors of the predictors as margins for band gap targets. Here, we have performed optimizations using the prediction errors shown in Table 1 as margins. As a result of the optimizations, the SMOACS still achieves the best results.

---

> > > > ### Comment · Reviewer_p9py · 2024-12-02
> > > >
> > > > Thank you for the clarifications. Your replies have addressed some of my concerns and I have therefore raised my score. However, I'm still not convinced that the numerical evaluation is really fair and it appears to be too tailored to the proposed method.

---

> ### Author Response · Authors · 2024-12-02
> **A clarification request for reviewer p9py**
>
> Thank you very much for your positive feedback and for raising our score. We sincerely appreciate your time and effort in reviewing our work.
>
> To further address your concern, we kindly ask for more detailed guidance.
>
> First, regarding the tuning of FTCP and TPE, as well as the comparison method with SMOACS, we have explained in "Response to your concerns and questions (1/4)" under "Weakness 1" that this constitutes a fair comparison.
>
> Second, concerning your apprehensions about the band gap width, we have clarified in "Response to your concerns and questions (3/4)" under "Weakness 3" that using a fixed band gap width was necessary for a fair comparison. Additionally, in "Add experiments using the prediction errors of the predictors as margins for band gap target," we have conducted experiments using the prediction errors of the models as margins for the band gap target, as per your suggestion, and included the results in Section A.14.
>
>  if there are any specific concerns that remain unresolved, we would appreciate it if you could kindly point them out.

---

> > ### Comment · Reviewer_p9py · 2024-12-03
> >
> > I still believe that the numerical evaluation is too tailored to the proposed method. I've read through the supplementary material, but it's still not clear to me how FTCP is set up to target the specified bandgap range (which is what results in a low success rate for this method). And if it's difficult to guide FTCP towards the desired properties, then perhaps it's not the most suitable generative method for the comparison? (see e.g. https://www.cell.com/matter/fulltext/S2590-2385(24)00242-X for a review of generative models that can be used for property -> structure generation)

---

> ### Author Response · Authors · 2024-12-04
> **Addressing your concern (1/2)**
>
> Thank you for your important comments. We will address your concerns by dividing our response into two parts: (1) the evaluation method of FTCP, and (2) the appropriateness of including FTCP as a comparative method.
> > (1) I still believe that the numerical evaluation is too tailored to the proposed method. I've read through the supplementary material, but it's still not clear to me how FTCP is set up to target the specified bandgap range (which is what results in a low success rate for this method).
>
> [__Reproduction of the methodology of the authors of FTCP__] First, we have followed the methods proposed by the authors of FTCP in our experiments, so your concern about the evaluation being "*too tailored to the proposed method*" does not apply. For example, the experiment in Section 4.2 is conducted exactly as described in Section A.6: “*FTCP selects initial data from the training dataset where the band gap is close to the target and the formation energy is below -0.5 eV, and then uses an encoder to convert this into latent variables. Next, we add noise to these latent variables using a normal distribution with a mean of 0 and a standard deviation of 0.6, then decode them back into crystal structures for evaluation.*” This procedure is consistent with the method described in the Workflow section of the FTCP paper. And as described Section 4.2 :“*We optimized and evaluated the structures based on the band gap and formation energy values predicted by their respective predictors* ” from latent variables of FTCP because “*It employs prediction branches to predict properties from these variables.*” as described in Section 2. Moreover, as stated in Section A.4, we used the official implementation, thereby fully adhering to the material design method with specified bandgap proposed by the FTCP authors.
>
> [__Simultaneous optimization of more targets__] Second, our paper proposes a novel problem of simultaneous multiple properties optimization. The number of property evaluation metrics in each paper is as follows: CDVAE (Section 5.3) evaluates one property, MatterGen evaluates two (e.g., crystal structure and energy (S.U.N)), FTCP evaluates four (bandgap, energy, electrical neutrality, and minimum interatomic distance), while our research evaluates six properties (bandgap, energy, electrical neutrality, preservation of crystal structure, tolerance factor, minimum interatomic distance). Therefore, conventional methods not designed to handle this complexity naturally achieve lower scores.
>
> In summary, we have strictly followed the original FTCP method without intentionally distorting the evaluation process. Although the scores for other methods may appear low, this is expected because we are tackling a new and more difficult problem.

---

> ### Author Response · Authors · 2024-12-04
> **Addressing your concern (2/2)**
>
> > (2) And if it's difficult to guide FTCP towards the desired properties, then perhaps it's not the most suitable generative method for the comparison? (see e.g. https://www.cell.com/matter/fulltext/S2590-2385(24)00242-X for a review of generative models that can be used for property -> structure generation)
>
> [__FTCP is a common benchmark__] Firstly, FTCP is a widely used benchmark in this field and is optimal for comparison. According to Google Scholar, FTCP has been cited 128 times, indicating its broad acceptance as a benchmark. It has also been used as a benchmark method in recent studies like MatterGen (Dec. 2023). Since we aim to compare our work with well-known methods in the field, FTCP is one of the most suitable choices.
>
> [__The number of evaluation metrics of FTCP is the most close to ours__] Secondly, FTCP is appropriate from the perspective of the number of evaluation metrics. As previously mentioned, we are solving a problem involving six property evaluations. FTCP evaluates four properties while CDVAE and MatterGen do one and three properties. So it is suitable to compare our method with others that assess a similar number of properties.
>
> [__All generative models share common limitations__] Thirdly, generative models are probabilistic models and share common limitations. Generative models such as FTCP generate materials probabilistically, and even when conditioned on specific crystal structures and electrical neutrality, they cannot guarantee these properties entirely. Additionary, they cannot apply adaptive constraints without retraining. Therefore, while using state-of-the-art generative models such UniMat or MatterGen might offer some performance improvement, they cannot fully address the challenges (also, UniMat and MatterGen's datasets and codes are not publicly available, making direct comparison difficult). In contrast, our method can guarantee these properties by restricting the optimization range and atomic distribution.It can also apply adaptive constraints without retraining. Consequently, similar results are expected regardless of which generative model is used, with some variations in scores.
>
> [__Success rate is competitive with the latest model__] In fact, although it is not a direct comparison, our success rate is not significantly different from MatterGen[3] even though we are tackling more challenging problems than they are. In Figure 4 of MatterGen, they perform energy minimization searches (S.U.N) under the condition of a given crystal structure (i.e., solving two conditional problems), addressing a total of 14 conditional problems related to crystal structures. However, out of these 14 cases, seven have success rates below 20%, with some even falling below 10% even though they use larger dataset and fine-tuning. In our experiments involving crystals having a similar number of atoms (Section 4.3), we achieved a success rate of approximately 10 ~ 20% under five conditions: three optimization targets, electrical neutrality, and preservation of the crystal structure. Therefore, compared to MatterGen, our success rate is by no means low.
> In conclusion, FTCP is a widely recognized benchmark method with a number of property evaluations similar to ours. Additionally, there is no fundamental reason to expect significant differences even if other methods are used. Therefore, FTCP is an optimal choice for comparison.

---

### Official Review · Reviewer_uNXA · 2024-11-02

**Soundness:** 2
**Presentation:** 1
**Contribution:** 1
**Rating:** 5
**Confidence:** 3

**Summary:**

This paper proposes SMOACS, which uses various property prediction models to perform backpropagation to simultaneously optimize the target properties. To maintain electrical neutrality, it restricts the possible values of atomic distribution. Experiments show that the method can achieve a good performance in simultaneously optimizing properties such as band gap and formation energy.

**Strengths:**

The problem of simultanously optimizing multiple properties for materials is important, and the paper proposes a method that seems to be effective.

**Weaknesses:**

Novelty: The method uses property prediction model and use backpropagation to optimize the properties. This kind of method has been employed in many prior works such as [1][2][3] and don't seem to be novel.

One limitation of the method is that it heavily relies on good property prediction models. In the case of not enough data, the result may not be good.

Clarity: The method needs to clearly state how the method differs from the prior methods (not just listing the prior methods). Also, it needs to be more clear when introducing the method (e.g., in the section 3.1).


[1] Allen, Kelsey R., et al. "Physical design using differentiable learned simulators." NeurIPS 2022

[2] Hwang, Rakhoon, et al. "Solving pde-constrained control problems using operator learning." Proceedings of the AAAI Conference on Artificial Intelligence. Vol. 36. No. 4. 2022.

[3] Zhao, Qingqing, David B. Lindell, and Gordon Wetzstein. "Learning to solve pde-constrained inverse problems with graph networks." ICML 2022

**Questions:**

N/A

---

> ### Author Response · Authors · 2024-11-20
> **Response to your concerns**
>
> We sincerely appreciate your valuable feedback regarding our manuscript. Your comments have provided significant guidance to enhance our research, and we are grateful for your thoughtful review.
>
> The latexdiff with the updated paper is available in the Supplementary Material at the following URL: https://openreview.net/attachment?id=NVKwjCIAAX&name=supplementary_material
>
> Below, we address the weaknesses you have raised.
>
>
>
> —---------
> > Weakness 1 : Novelty: The method uses property prediction model and use backpropagation to optimize the properties. This kind of method has been employed in many prior works such as [1][2][3] and don't seem to be novel.
>
> As you mentioned, employing backpropagation to optimize properties is indeed well-established, with applications across dynamics, image manipulation[4], and nanoscale material design[5]. However, to the best of our knowledge, our approach, which involves directly optimizing crystal structures via backpropagation, appears to be unprecedented. A significant aspect of our work is transforming non-differentiable features of crystal structures—such as crystal types and elemental species—into a differentiable format, which is a contribution in this area. Although Konno et al. [6] developed the transformation of compositions, we apply this method to incorporate site-specific constraints on atomic probabilities for gradient-based optimizations, which constitutes our novel contribution.  Your comments have highlighted the necessity of situating our work within the broader context of existing literature.
> To address this, we have introduced a comprehensive discussion in the related works section under "Gradient-Based Optimization."  Additionally, we have now included a new paragraph to show the novelty in Introduction as follows:
> "SMOACS is the first method that directly optimizes the space of crystal structures using a gradient-based approach. We achieve this by making the entire crystal structure differentiable, which involves decomposing it into various components and representing atomic species as atomic distributions. Unlike traditional methods that convert crystal structures into latent variables—thereby entangling their elements—our approach maintains the independence of each component. This independence facilitates the preservation of crystal structures and ensures electrical neutrality by precisely specifying the atoms at each site. Furthermore, unlike generative models that probabilistically generate materials satisfying certain conditions, our method can fundamentally guarantee electrical neutrality and the preservation of crystal structures. Moreover we can add additional constraints as long as they are differentiable."
>
>
> —---------
> > Weakness 2 : One limitation of the method is that it heavily relies on good property prediction models. In the case of not enough data, the result may not be good.
>
> Thank you for highlighting the dependency on the accuracy of property prediction models. Indeed, if a model lacks predictive accuracy, it becomes challenging to propose high-quality materials. Nevertheless, a key advantage of our approach is its capacity to leverage various high-performance models. This allows us to propose materials for crystal structures with sparse data—such as perovskites—by utilizing models that have been trained on larger datasets. We demonstrate this ability in Section 4.3, where we effectively optimize perovskite structures using the widely accessible the ALIGNN and Crystalformer models, which has been trained on the MEGNet dataset including various crystal structures. This example underscores our method's adaptability and effectiveness even with limited data scenarios.
>
> —---------
> > Weakness 3 : Clarity: The method needs to clearly state how the method differs from the prior methods (not just listing the prior methods). Also, it needs to be more clear when introducing the method (e.g., in the section 3.1).
>
> Thank you for your critical feedback. To address your concerns, we have clearly outlined this by adding a new paragraph in the Introduction section as described in Weakness 1.
>
> —------
>
> I hope that all of your concerns are resolved. Thank you very much for your attention!
>
>
> [4]  Xia, Weihao, et al. "Gan inversion: A survey." IEEE transactions on pattern analysis and machine intelligence 45.3 (2022): 3121-3138.
>
> [5] Ren, Simiao, Willie Padilla, and Jordan Malof. "Benchmarking deep inverse models over time, and the neural-adjoint method." Advances in Neural Information Processing Systems 33 (2020): 38-48.
>
> [6]  Konno, Tomohiko, et al. "Deep learning model for finding new superconductors." Physical Review B 103.1 (2021): 014509.

---

> > ### Comment · Reviewer_uNXA · 2024-12-01
> > **Response for the rebuttal**
> >
> > Thank you for the response. I appreciate the clarification on the novelty, related work and the limitations the authors made.Yet, I think current diffusion-based approaches need to be compared against, to demonstrate the method's effectiveness.
> >
> > Based on the above, I have raised my score to 5, and hope that the authors incorporate these comparisons in their paper.

---

> ### Author Response · Authors · 2024-12-04
> **Comparison with the latest diffusion models**
>
> Thank you very much for your valuable feedback and for raising our score.
>
> [__The latest diffusion-based models are not publicly available__] Firstly, we would like to point out that comparing our method with the latest diffusion-based models is challenging due to their unavailability. As mentioned in the "Deep Generative Models" section of Related Works, there are limited studies that address both desired material properties and structural stability simultaneously, which is the focus of our research. Among these, UniMat[1] and MatterGen[2] are the most recent efforts utilizing diffusion models for this purpose. However, neither their datasets nor codes are publicly accessible, making it impossible to conduct direct comparisons. (Although UniMat has a GitHub page, the code is not provided.)
>
> [__The number of evaluation metrics of FTCP is the most close to ours__] Secondly, FTCP is more appropriate from the perspective of the number of evaluation metrics than MatterGen and UniMat. The number of property evaluation metrics in each paper is as follows: CDVAE (Section 5.3) evaluates one property, MatterGen and UniMat evaluate two (e.g., crystal structure and energy (S.U.N)), FTCP evaluates three (bandgap, energy, electrical neutrality, and minimum interatomic distance), while our research evaluates six properties (bandgap, energy, electrical neutrality, preservation of crystal structure, tolerance factor, minimum interatomic distance).  So it is suitable to compare our method with FTCP that assesses the most similar number of properties.
>
> [__Success rate is competitive with the latest models__] Thirdly, although it is not a direct comparison, our success rate is not significantly different from MatterGen[3] even though we are tackling more challenging problems than they are. In Figure 4 of MatterGen, they perform energy minimization searches (S.U.N) under the condition of a given crystal structure (i.e., solving two conditional problems), addressing a total of 14 conditional problems related to crystal structures. However, out of these 14 cases, seven have success rates below 20%, with some even falling below 10% even though they use larger dataset and fine-tuning. In our experiments involving crystals having a similar number of atoms (Section 4.3), we achieved a success rate of approximately 10 ~ 20% under five conditions: three optimization targets, electrical neutrality, and preservation of the crystal structure. Therefore, compared to MatterGen, our success rate is by no means low.
>
> [__All generative models share the same limitations__] Fourthly, we believe that FTCP serves as a suitable substitute for the latest diffusion models in our comparisons. Generative models, including FTCP (VAEs) and the latest diffusion models, generate crystals probabilistically and cannot guarantee electrical neutrality or specific crystal structures and cannot apply adaptive constraint without retraining. This fundamental limitation remains unchanged even with the latest diffusion models. In contrast, our method can ensure those. Our objective is to demonstrate that, unlike these generative models, our method can maintain electrical neutrality and specific crystal structures without retraining, as shown in Table 3.
> To summarize, FTCP is more appropriate from the perspective of the number of evaluation metrics. Additionally, FTCP is sufficient as a representative of generative models—including the latest diffusion models—and the latest diffusion models cannot be utilized because their data is not publicly available. Additionally, although it is not a direct comparison, our success rate is not significantly different from MatterGen.
>
> We hope this addresses your concerns and will consider incorporating these comparisons into our revised manuscript.
>
> [1] Yang, Sherry, et al. "Scalable diffusion for materials generation." arXiv preprint arXiv:2311.09235 (2023).
>
> [2] Zeni, Claudio, et al. "Mattergen: a generative model for inorganic materials design." arXiv preprint arXiv:2312.03687 (2023).

---

### Official Review · Reviewer_v3Du · 2024-11-03

**Soundness:** 3
**Presentation:** 3
**Contribution:** 3
**Rating:** 6
**Confidence:** 4

**Summary:**

The paper proposes a new method for multi-property optimization of materials called Simultaneous Multi-property Optimization using Adaptive Crystal Synthesizer (SMOACS). SMOACS uses the gradients from neural network based property prediction to optimize materials structures within preset constraints. The paper first introduces the motivation for multi-property materials design that can be constrained for certain types of materials, such as perovskites. The paper also details the important of charge neutrality in materials design, which later becomes a consideration for the proposed design formulation. The introduction also describes past advances in related materials generation methods, such as generative models and bayesian optimization. In Section 2, the paper describes related works focusing on property prediction models, deep generative models and bayesian optimization providing further context for the introduction.

Section 3 of the paper describes the relevant details for SMOACS, including the concrete details of the how the crystal design is constructed along lattice constants and angles, oxidation state, atomic sites and elements. To maintain charge neutrality, the paper proposes a masking method to restrict the actions related to the oxidation state which is described in Section 3.1. Section 3.2 outlines the details of multi-property optimization using training signals from neural network based property prediction models and Section 3.3 describes how SMOACS maintains predefined crystal structures by limiting the range of certain design variables.

Section 4 details the experiments of the paper, including the property prediction models used for the SMOACS algorithm (ALIGNN and Crystalformer). Section 4.2 outlines an experiments without constraints on the crystal structure and Section 4.3 details an experiment with crystal structures constrained to perovskites. In both cases, SMOACS shows general outperformance compared to the Bayesian optimization-based baselines. In addition to the design experiments described in the paper, the authors perform DFT verification of the band gaps of two of the materials designed by SMOACS which showed some agreements and discrepancy. Section 5 consists of a conclusion that summarizes the main findings of the paper

**Strengths:**

* The paper presents a new method for materials design that effectively utilizes the gradients from property prediction models to optimize materials designs.
* The ability to condition materials design on different properties is a useful and novel feature that many generative models today fail to achieve and as such provide an advantage for SMOACS.
* SMOACS can also enforce constraints through the range of the design values, which could be useful to a variety of design cases.

**Weaknesses:**

* The experiments to just one dataset, which means the scope is a bit limited. The authors should consider running further experiments based on crystal structure datasets [1], as well as Perov-5 for perovskites [2].
* The paper could be strengthened by adding more experiments with additional property prediction models, such as commonly used machine learning potentials [3] which should also be covered in the related work section. In the case that some models cannot be used (e.g., because they cannot predict relevant properties) this should also be explained.
* In terms of related work, the authors could also benefit from discussing the application of reinforcement learning [4] and GFlowNets [5] for crystal structure design. The same should also be applied to diffusion models [6], flow matching models [7] and language models [8] [9]. It would be good to add details about how those methods compare.

[1] Lee, Kin Long Kelvin, et al. "Matsciml: A broad, multi-task benchmark for solid-state materials modeling." arXiv preprint arXiv:2309.05934 (2023).

[2] Castelli, Ivano E., et al. "New cubic perovskites for one-and two-photon water splitting using the computational materials repository." Energy & Environmental Science 5.10 (2012): 9034-9043.

[3] Bihani, Vaibhav, et al. "EGraFFBench: evaluation of equivariant graph neural network force fields for atomistic simulations." Digital Discovery 3.4 (2024): 759-768.

[4] Govindarajan, Prashant, et al. "Learning conditional policies for crystal design using offline reinforcement learning." Digital Discovery 3.4 (2024): 769-785.

[5] AI4Science, Mila, et al. "Crystal-gfn: sampling crystals with desirable properties and constraints." arXiv preprint arXiv:2310.04925 (2023).

[6] Jiao, Rui, et al. "Crystal structure prediction by joint equivariant diffusion." Advances in Neural Information Processing Systems 36 (2024).

[7] Miller, Benjamin Kurt, et al. "FlowMM: Generating Materials with Riemannian Flow Matching." Forty-first International Conference on Machine Learning.

[8] Gruver, Nate, et al. "Fine-Tuned Language Models Generate Stable Inorganic Materials as Text." The Twelfth International Conference on Learning Representations.

[9] Ding, Qianggang, Santiago Miret, and Bang Liu. "MatExpert: Decomposing Materials Discovery by Mimicking Human Experts." arXiv preprint arXiv:2410.21317 (2024).

**Questions:**

* How are you obtaining gradient signals for properties that are not part of the property prediction model, such as oxidation state?
   * Does SMOACS have limitations when input variables or properties are not covered by the prediction model?
* Can you add more details on the compute cost and infrastructure used to train SMOACS and how it compares to the baselines studied in the paper?
* While SMOACS performs better than the bayesian optimization baselines, the success rate is still low in many cases. Can you provide some intuition as to why and how future work could improve on this?

---

> ### Author Response · Authors · 2024-11-20
> **Response to your concerns and questions (1/2)**
>
> We sincerely appreciate your valuable feedback and insightful questions regarding our manuscript. Your comments have provided significant guidance to enhance our research, and we are grateful for your thoughtful review.
>
> The latexdiff with the updated paper is available in the Supplementary Material at the following URL: https://openreview.net/attachment?id=NVKwjCIAAX&name=supplementary_material
>
> Below, we address the weaknesses and questions you have raised.
>
> > weakness1 : The experiments to just one dataset, which means the scope is a bit limited. The authors should consider running further experiments based on crystal structure datasets [1], as well as Perov-5 for perovskites [2].
>
>
> One of the main strengths of our approach is its ability to maintain specific crystal structures during optimization using any pretrained model. Consequently, the dataset models trained are less crucial in our method. What is more significant is verifying that our technique can be applied effectively across diverse types of models. To validate this, we have conducted experiments with two web-available pretraineds: a GNN model (ALIGNN) and a Transformer model (Crystalformer). We demonstrated that our method can successfully optimize using both models.
> On the other hand, we understand your concerns. We are currently investigating whether we can conduct experiments using models trained on other datasets.
>
>
>
> > weakness 2 : The paper could be strengthened by adding more experiments with additional property prediction models, such as commonly used machine learning potentials [3] which should also be covered in the related work section. In the case that some models cannot be used (e.g., because they cannot predict relevant properties) this should also be explained.
>
> Thank you for your valuable suggestions. After careful consideration, we have concluded that machine learning potentials serve different purposes from our research objectives.  Machine learning potentials are primarily designed to predict energies and forces but typically do not extend to electronic properties like the band gap. Therefore, incorporating these models into our study may not align with our research objectives.
>
> The number of factors being explored, our research is on par with other studies. MatterGen explores five factors: stability (S.U.N.), magnetism, limitation of atomic species (composition formula and supply chain risks), and 14 types of crystal structures (space symmetry). In our paper, we also explore five factors: in addition to stability, we consider eight types of band gaps (Table A.3, A.4, and A.5), crystal structures, tolerance factor, and electrical neutrality. Although the tolerance factor is not a value through a prediction model, please note that it is an optimization target as shown in Equation (17). Therefore, in Sections 4.3 and 4.4, we optimize three properties.
> Based on the above discussions, we believe that the extensiveness of our experiments is sufficient.
>
> —----
> > weakness 3 : In terms of related work, the authors could also benefit from discussing the application of reinforcement learning [4] and GFlowNets [5] for crystal structure design. The same should also be applied to diffusion models [6], flow matching models [7] and language models [8] [9]. It would be good to add details about how those methods compare.
>
> Thank you for your critical insights. Each of the suggested papers seems very important in the field of material design. We appreciate this opportunity to expand our discussion. We have updated the introduction and related work sections to include these papers.
>
> However, we believe that Reference [6], which addresses CSP (Crystal Structure Prediction) to search for appropriate crystal structures from compositions, has a somewhat different objective from the material design we are conducting in the main text. Therefore, we have added a discussion in Section A.3 where we have done a similar experiment.

---

> ### Author Response · Authors · 2024-11-20
> **Response to your questions (2/2)**
>
> > Question 1 : How are you obtaining gradient signals for properties that are not part of the property prediction model, such as oxidation state?
>
> Regarding the oxidation state, we do not utilize gradient signals to achieve electrical neutrality. Instead, we first determine D patterns of oxidation states that achieve electrical neutrality, as explained in Section 3.1. (e.g. [+2,−1,−1] and  [+4,−2,−2] in Section 3.1.) We then employ the method illustrated in Figure 2 to create masks for each atomic site, thereby restricting the possible elements that can occupy each site. This ensures that the optimized materials are electrically neutral. Regardless of the optimization process's success or failure, electrical neutrality is inherently achieved through the predefined oxidation state patterns and the masking technique. Additionally, we realized that the procedure for obtaining the D patterns of oxidation states that achieve electrical neutrality might not have been sufficiently clear. To address this, we have enhanced the following sentence in Section 3.2 to clarify the 'D':
> In SMOACS, optimizations begin with crystal structures from a dataset or those randomly generated. These crystal structures must satisfy electrical neutrality and generate D of oxidation number patterns based on the compositions of initial crystal structures (see Section A.11 for details).
>
> In our framework, we rely on gradient signals for optimization unless we implement specific strategies, such as the approach we used for oxidation states. For example, in Section 4.3, we optimize the tolerance factor because it is expressed through differentiable computations, enabling the use of gradient-based methods. However, certain properties pose challenges. The number of atoms in the initial structure cannot be described in a differentiable form, which means it cannot be optimized using gradient signals. As a result, the initial structure and the number of atoms it contains must be set as hyperparameters rather than optimized variables.
>
> —---------
> > Question 2 : Can you add more details on the compute cost and infrastructure used to train SMOACS and how it compares to the baselines studied in the paper?
>
> Thank you for your insightful question. We have compiled the software information and included it in the implementation details of Section A.4. We recognize the importance of providing details about the computational infrastructure used to train SMOACS, including information about GPUs and other resources. We have included this information in section A.4.
>
> We did not include a comparison of the computational costs between SMOACS, TPE, and FTCP in the original manuscript because making a fair comparison is challenging due to constraints in our current computational setup. Specifically, SMOACS is implemented using PyTorch, whereas the official implementation of FTCP utilizes TensorFlow 1.x. TPE primarily runs on CPUs, but the component responsible for obtaining predicted values employs GPUs, and implementing parallelization for TPE is not straightforward. Due to these differences and the complexities involved in optimizing each method under varying hardware conditions, we have not conducted measurement experiments for a direct comparison.
>
> Nevertheless, although we did not perform a formal comparison, we might say that SMOACS operates entirely on GPUs, which may be easy to facilitate more efficient generation of a larger number of samples compared to TPE. For example, using CrystalFormer, we can simultaneously optimize 2,048 crystal structures containing five atoms each, and the computation completes in a matter of minutes.
>
> —-------
> > Question 3 : While SMOACS performs better than the bayesian optimization baselines, the success rate is still low in many cases. Can you provide some intuition as to why and how future work could improve on this?
>
> Thank you for your thoughtful question. The reason the success rate is not higher could be primarily due to the difficulty in overcoming local minima. Gradient descent algorithms can be trapped in local optima, which limits the overall success rate in certain cases. However, we do not consider the low success rate to be a significant issue. SMOACS is highly efficient; using a single A100 GPU, we can optimize 2,048 samples in a matter of minutes. This allows us to repeat the optimization process multiple times, enabling us to obtain a large number of successful optimization samples. Your comments have underscored the importance of incorporating this information into the main body of our text. Consequently, we have added the following sentences:
>
> “SMAOCS can easily scale this computation and can optimize 2,048 samples simultaneously in just a few minutes using a single A100 GPU. This allows us to repeat the optimization process multiple times, enabling us to obtain a large number of successful optimization samples. “ in Section 4.1
>
> —-------
>
> I hope that all of your concerns are resolved. Thank you.

---

> > ### Author Response · Authors · 2024-11-22
> > **Additional experiment regarding to Weakness1**
> >
> > > Weakness1: The experiments to just one dataset, which means the scope is a bit limited. The authors should consider running further experiments based on crystal structure datasets [1], as well as Perov-5 for perovskites [2].
> >
> > Thank you for your insightful feedback regarding the scope of our experiments. In response to your suggestion, we have expanded our study to include two additional datasets from JARVIS[10]: the JARVIS DFT dataset and JARVIS Supercon. To evaluate properties distinct from the electronic property (band gap), we conducted experiments using these datasets, which facilitate the optimization of mechanical properties (bulk modulus) and superconducting transition temperatures. We used ALIGNN models trained on each of these datasets and conducted comprehensive experiments, which are summarized in Section A.13. Although the success rate appears lower, we do not regard this as a significant concern. SMOACS demonstrates high efficiency, capable of optimizing thousands of samples within a few ~ 30 minutes using a single A100 GPU, as detailed in Section 4.1.
> >
> > With the inclusion of these datasets, our analysis now encompasses three datasets: MEGNet, JARVIS DFT, and JARVIS Supercon, thereby broadening the applicability and robustness of our findings. We add the following sentence to the first paragraph of Section 4 to ensure readers are aware: “For experiments utilizing models trained on datasets other than MEGNet, please refer to Section A.13.”
> >
> > [10] Choudhary, K. et al. The joint automated repository for various integrated simulations (JARVIS) for data-driven materials design. npj Computational Materials, 6(1), 1-13 (2020).

---

> > > ### Comment · Reviewer_v3Du · 2024-11-22
> > > **Property Prediction Models**
> > >
> > > Some additional questions:
> > >
> > > - Could you clarify if you need a model trained exactly on the property to apply SMOACS? If so, would it be possible to fine-tune a machine learning potential model toward that property?
> > > - Why is the low success rate not a significant issue?

---

> ### Author Response · Authors · 2024-11-24
> **Response to your questions about prediction models and success rate**
>
> Thank you for your questions. We are pleased to provide clarifications below.
>
>
> > Question 1: Could you clarify if you need a model trained exactly on the property to apply SMOACS? If so, would it be possible to fine-tune a machine learning potential model toward that property?
>
> Yes. Optimizing crystal structures with SMOACS to achieve a specific property requires a predictive model for that property. We acknowledge that fine-tuning machine learning potential (MLP) models toward the property of interest (e.g., band gap) should, in principle, allow applying SMOACS. (Of course, in this case, differentiable connections from input to output are also necessary.)
>
> However, we are afraid that such fine-tuning for applying to SMOACS may not provide expected good performance because of their architecture optimized for prediction of energy and forces. For example, in all state-of-the-art MLP (e.g. [11],  [12]) models, the total energy is expressed as a sum of atomic energy to attain transferability to systems having different number of atoms from those in the training dataset. Obviously, when predicting the band gap or Tc, such a form, that is, decomposition into individual atomic contributions, is unreasonable.  Models like ALIGNN and Crystalformer, which do not decompose properties into atomic-level contributions, might be more appropriate for these tasks.
>
> For this reason, we limited the models used in our experiments to those with proven performance in predicting properties such as band gap, formation energy, and bulk modulus, such as ALIGNN and Crystalformer.
>
> > Question 2: Why is the low success rate not a significant issue?
> SMOACS can optimize thousands of samples in a few minutes to about 30 minutes on a single A100 GPU. By repeating the optimization or running it in parallel on multiple GPUs, we can obtain dozens to hundreds of successful samples within an hour, even with a success rate around 10%.
>
> SMOACS can optimize thousands of samples in a few minutes to about 30 minutes on a single A100 GPU. By repeating the optimization or running it in parallel on multiple GPUs, we can obtain dozens to hundreds of successful samples within an hour, even with a success rate around 10%.
> In materials design, candidates usually undergo time-consuming processes like DFT calculations or experimental synthesis. Since SMOACS generates promising candidates much faster than these steps, the low success rate doesn’t hinder the overall efficiency of material discovery. Therefore, the success rate isn’t a significant concern in practical applications.
>
> While similar issues might apply to FTCP and TPE, their situations differ from SMOACS. In TPE, we must check electrical neutrality after optimization, even if all specified properties are met. This becomes serious when proposing large systems, as discussed in Section 4.4. FTCP requires extra verification to ensure both electrical neutrality and the specified crystal structure, adding complexity. In contrast, SMOACS inherently ensures electrical neutrality and the specified crystal structure, eliminating these concerns.Note that for the sample evaluation speed description, it is detailed in Section 4.2, not Section 4.1, as previously mentioned.
>
>
> We hope the above addresses your questions and provides a clearer understanding of our methodology. Thank you again for your valuable feedback.
>
> [11] Musaelian, A., Batzner, S., Johansson, A. et al. Learning local equivariant representations for large-scale atomistic dynamics. Nat Commun 14, 579 (2023). https://doi.org/10.1038/s41467-023-36329-y
>
> [12] Park, Yutack, et al. “Scalable Parallel Algorithm for Graph Neural Network Interatomic Potentials in Molecular Dynamics Simulations.” Journal of Chemical Theory and Computation (2024)

---

### Author Response · Authors · 2024-12-04
**General comments (1/3)**

We sincerely thank all reviewers for their encouraging and insightful comments. In this study, we propose a new method, SMOACS, which can obtain crystal structures that simultaneously achieve multiple target properties by optimizing the input (crystal structures) using off-the-shelf pretrained models and their gradients.

As highlighted by the reviewers:
- Our work "*proposes a novel problem of simultaneous multiple properties optimization*" (__stg3__) and "*The problem of simultaneously optimizing multiple properties for materials is important*" (__uNXA__).
- We propose "*a method that seems to be effective*" (__uNXA__) and "*The ability to condition materials design on different properties is a useful and novel feature that many generative models today fail to achieve and as such provide an advantage for SMOACS*" (__v3Du__).
- Our method "*makes it easy to constrain the optimization algorithm to respect e.g. structure constraints by simply turning off the optimization of some variables*" (__p9py__) and "*SMOACS can also enforce constraints through the range of the design values, which could be useful to a variety of design cases*" (__v3Du__).

Currently, all revisions are reflected in the main paper, and the latexdiff with the first manuscript is available in the [Supplementary Material](https://openreview.net/attachment?id=NVKwjCIAAX&name=supplementary_material). Here are the main modifications:

- (New) Section A.12 Adjustment of Priorities in the Loss Function
- (New) Section A.13 Experiments with models trained on other datasets
- (New) Section A.14 Optimization with Margins Based on Predictor Error
- Two paragraphs are added in the Introduction section: the first one describes a summary early on the paper; the second one describes the novelty and strength of our method.

As shown below, we have sincerely addressed or clearly rebutted all the concerns of the reviewers. Therefore, we believe that __all concerns have been resolved__.

### Comments on Suggesting Additional Experiments
- __stg3__ pointed out the influence of the weighting parameter $\lambda$ when optimizing multiple targets. We had adopted $\lambda=1$ for a fair comparison, but since it is possible to adjust $\lambda$ in practice, we investigated the effect when varying $\lambda$ and added the results to Section A.12 "Adjustment of Priorities in the Loss Function."

- __v3Du__ and __stg3__ noted that models used in our method were only trained on MEGNet or that the types of properties being optimized were few. To address this, we prepared a total of three models trained on two types of datasets and additionally optimized three properties and added this to Section A.13 “Experiments with models trained on other datasets”. As a result, the datasets we verified increased to three, and the properties optimized increased to six types.

- __v3Du__ suggested including experiments using machine learning potentials (MLPs). However, since MLPs are models specialized in predicting energy and forces, it is difficult to extend them to other properties such as bandgap. Therefore, we responded that this does not align with our objectives.

- __p9py__ proposed using the error of the predictive model as the optimization margin, rather than using a fixed margin (e.g., 0.04 eV) for the target range of optimization (e.g., bandgap: 4.00 ± 0.04 eV). We had used fixed margin for fair comparison but acknowledged that such a method may be used in practical settings. We included experimental results in Section A.14 "Optimization with Margins Based on Predictor Error." As a result, we demonstrated that even in this case, the proposed method outperforms other methods.


### Novelty
__uNXA__ pointed out that there are other methods that optimize inputs using gradients and that the differences between them and our method are unclear. __stg3__ commented to clarify the differences from the generative model CDVAE[1] that uses gradients. We responded that the research referenced by __uNXA__ deals with fluids, whereas our method is the first method to optimize directly within the space of crystal structures, making it fundamentally different. Additionally, CDVAE is a generative model that encodes crystal structures into latent space and optimizes within that space. Therefore, it is fundamentally different from our method, which allows for easy application of various conditioning by optimizing directly in the space of crystal structures rather than the latent space. Furthermore, unlike CDVAE, which requires training a generative model, our method can perform optimization using only off-the-shelf pretrained models, highlighting that our approach is entirely different in framework from generative models. To convey these points more clearly, we improved our paper as follows: We explicitly stated the relation to other methods using gradients in the related works and added a new section in the Introduction that describes the differences and strengths compared to related methods.

---

> ### Author Response · Authors · 2024-12-04
> **General comments (2/3)**
>
> ### Related Works and Writing
> - __v3Du__ presented several studies closely related to our research. We strengthened our paper by incorporating them into the Introduction and Related Works sections.
>
> - __p9py__ pointed out the lack of a concise formulation of the problem and description of the proposed method early in the paper. We revised our manuscript by adding a new first paragraph to the Introduction to clearly state these points.
>
> - __p9py__ highlighted the potential application of training-free conditional generative models to crystal structure generation. We have stated that no such training-free conditional generative models exist for crystal structure generation and have added the presented research to the related works section.
>
> ### Limitations of the Proposed Method
> - __uNXA__ indicated that our method heavily relies on the accuracy of pretrained models, and if the accuracy of these models is low, good material proposals cannot be made. However, as __p9py__ pointed out that it is "straightforward to use any available predictive model," we stated that this can be resolved by using high-quality off-the-shelf models. Considering that new and more accurate property prediction models are published at ICLR every year, it is reasonable to expect that our method's will be able to use more accurate models.
>
> - __stg3__ expressed concern that the success rate is "poor." However, detailed justifications for this assessment were not provided. We pointed out that SMOACS is solving problems that previous studies could not address, as __p9py__ pointed out. Furthermore, we provided evidence that MatterGen [2] and CDVAE [1], which tackle problems with fewer optimization targets than ours, have achieved similar scores to our results. Additionally, we demonstrated that obtaining 500 successful samples in 1 GPU-hour is sufficiently practical for material design. Moreover, as __stg3__ himself (or herself) pointed out, since we are considering a new problem setting in an unknown domain where solutions may not even exist, we argued that it is natural for the success rate to be lower than standard machine learning benchmark scores. Thus, the success rate is sufficient rather than ‘poor’.
>
> - __stg3__ also expressed concern that randomly initializing the crystal structure is not a good way to overcome local minima. However, detailed justifications for this assessment were not provided. We answered that our initialization method is not entirely random; it is based on physics domain knowledge. In SMOACS, the initial structures for the optimization process are obtained from two sources: an existing dataset and self-generated structures based on typical perovskites. We mimic common techniques in physics such as element substitution. From a condensed matter physics perspective, although the crystal structure search space appears vast, practical materials occupy limited regions. Given this, our strategy of starting from known crystal structures and optimizing locally is the most efficient way to optimize crystals rather than using a good optimization technique to overcome local minima. We have included the details of the initialization in the main text of the manuscript and added a reference to the Appendix.
>
> - __p9py__ noted that, compared to generative models, there is a limitation in our method's ability to generate stable materials (being in a more favorable energy state than competing phases) and suggested that this should be discussed in the paper. However, MatterGen's paper [2] reveals that only a relatively small proportion of materials generated by the generative model were actually the most stable. From this perspective, accounting for the practical stability of materials remains a future challenge for both SMOACS and generative models.

---

> ### Author Response · Authors · 2024-12-04
> **General comments (3/3)**
>
> ### Comparison with other methods
> - __p9py__ expressed concern that the evaluation method of FTCP used in our comparison might be "*too tailored to the proposed.*" As we have thoroughly described in the main text and Appendix, our evaluation method strictly follows the methods of the FTCP authors, including the use of their code, and thus is not "*too tailored to the proposed.*" Furthermore, as __stg3__ pointed out, our paper "*proposes a novel problem of simultaneous multiple properties optimization*" and we need to recognize that we are addressing a problem that is challenging for conventional material design methods. The scores for FTCP may appear low, but this is expected because we are tackling a more difficult problem.
>
> - __uNXA__ suggested comparing with the latest diffusion models. Additionally, __p9py__ is concerned about whether there are other suitable generative models besides FTCP. We explain as follows: FTCP is the most appropriate from the perspective of the number of property evaluation (optimization) metrics. Additionally, FTCP is sufficient as a representative of generative models, including the latest diffusion models, because they share the same inherent limitations of generative models. Moreover, the latest diffusion models such as MatterGen cannot be utilized because their data and code are not publicly available. Furthermore, although it is not a direct comparison, our success rate is not significantly different from that of the latest diffusion model, MatterGen.
>
>
> ### Others
> All the points raised here are important, but they were explicitly stated in the Appendix of the initial manuscript.
>
> - __p9py__ pointed out that the implementation is unclear and that it is not evident whether a fair comparison is being made. We explained that we have already detailed this in the Appendix and that it constitutes a fair comparison.
>
> - __p9py__ noted that updating the graph data during optimization is necessary. We indicated that this is described in the Appendix A.4.
>
> [1] Tian Xie, ICLR2022, “Crystal Diffusion Variational Autoencoder for Periodic Material Generation”
>
> [2] Zeni, Claudio, et al. "Mattergen: a generative model for inorganic materials design." arXiv preprint arXiv:2312.03687 (2023).

---

### Meta-Review · Area_Chair_4ZYp · 2024-12-23

**Metareview:**

Summary
=======
The paper presents SMOACS, a method for multi-objective optimization of crystal structures. The key innovation is using gradients from neural network property predictors to directly optimize material designs while maintaining constraints like charge neutrality. The method allows for conditional generation with property constraints and shows better performance than Bayesian optimization baselines in experiments optimizing properties like band gap and formation energy. Some DFT verification was performed showing mixed agreement with predictions.

Strengths
=======
* Multi-objective optimization under constraints is an important problem in materials design
* No retraining is needed, as expected from optimization algorithms
* Supports explicit constraints on crystal structures through design variable restrictions

Weaknesses
=========
* Limited novelty, as gradient-based optimization using property predictors has been demonstrated in previous work
* Experiments are limited in scope (one dataset, two properties) and may not represent a fair comparison to baselines
* The method struggles to ensure stability/synthesizability of generated materials compared to generative approaches
* Success rates are relatively low
* The clarity of the methodology, particularly in describing how constraints and optimization parameters are handled, could be improved.
* Key issues such as the selection of templates for oxidation states, handling of model non-differentiability, and stability considerations remain underexplored.


Reasons for decision
================
The core technical approach appears to be incremental rather than novel in the machine learning literature, with similar gradient-based optimization methods already established. There are other major weaknesses as identified above.

**Additional Comments On Reviewer Discussion:**

The discussions were engaging and the authors have attempted to address many concerns raised. The scores where raised accordingly.

---

### Decision · Program_Chairs · 2025-01-22

Reject